# Heterogeneity estimates in a biased world

**Johannes Hönekopp** *, **Audrey Helen Linden**

Department of Psychology, Northumbria University, Newcastle upon Tyne, United Kingdom

* johannes.honekopp@unn.ac.uk

## Abstract

Meta-analyses typically quantify heterogeneity of results, thus providing information about the consistency of the investigated effect across studies. Numerous heterogeneity estimators have been devised. Past evaluations of their performance typically presumed lack of bias in the set of studies being meta-analysed, which is often unrealistic. The present study used computer simulations to evaluate five heterogeneity estimators under a range of research conditions broadly representative of meta-analyses in psychology, with the aim to assess the impact of biases in sets of primary studies on estimates of both mean effect size and heterogeneity in meta-analyses of continuous outcome measures. To this end, six orthogonal design factors were manipulated: Strength of publication bias; 1-tailed vs. 2-tailed publication bias; prevalence of $p$-hacking; true heterogeneity of the effect studied; true average size of the studied effect; and number of studies per meta-analysis. Our results showed that biases in sets of primary studies caused much greater problems for the estimation of effect size than for the estimation of heterogeneity. For the latter, estimation bias remained small or moderate under most circumstances. Effect size estimations remained virtually unaffected by the choice of heterogeneity estimator. For heterogeneity estimates, however, relevant differences emerged. For unbiased primary studies, the REML estimator and (to a lesser extent) the Paule-Mandel performed well in terms of bias and variance. In biased sets of primary studies however, the Paule-Mandel estimator performed poorly, whereas the DerSimonian-Laird estimator and (to a slightly lesser extent) the REML estimator performed well. The complexity of results notwithstanding, we suggest that the REML estimator remains a good choice for meta-analyses of continuous outcome measures across varied circumstances.

## Introduction

Meta-analyses pool the results from pertinent primary studies to estimate the magnitude and heterogeneity of the phenomenon under investigation. Typically, the results from individual studies vary more strongly than expected from sampling variance alone, which points to heterogeneity [1, 2]. As an example consider the sex difference in students' math performance, for which an international survey found substantial variation; e.g. boys did considerably better than girls in Italy, but the reverse pattern was observed in Saudi Arabia [3]. Being larger than expected from sampling error, this variability between countries is an example of

**Data Availability Statement:** All materials and data can be found at https://osf.io/qga8v/.

**Funding:** The authors received no specific funding for this work.

**Competing interests:** The authors have declared that no competing interests exist.

heterogeneity. Meta-analyses can quantify heterogeneity and thereby provide important information about the stability of the studied effect across contexts, i.e. different populations, times, research methods, etc. [4]. They can also try to uncover where heterogeneity comes from. For example, trans-national variability in the sex difference in students' math performance is (partly) explained by national differences in women's career opportunities [5].

A number of heterogeneity estimators have been proposed [6, 7]. Computer simulations that evaluate their performance typically presume that the set of primary studies underpinning the meta-analysis provides unbiased estimates of the underlying population effect size. In many contexts, this might be unrealistic [8, 9]. Whereas the effect of bias in sets of primary studies on meta-analytic effect size estimates has received considerable attention, its effect on heterogeneity estimates is less well understood [10–12]. Here, we report computer simulations that compare the performance of different heterogeneity estimators when applied to unbiased and biased sets of primary studies. We also compare how bias in the set of primary studies affects estimates of mean effect size and heterogeneity.

Our paper is organized as follows. In the introduction, we first address why heterogeneity matters before we deal with biases in sets of primary studies and what is known about their effects on meta-analysis. This motivates a more detailed account of our aims. In the methods section, we deal with the random effects model and the heterogeneity estimators that underpin our simulation before we address it in detail.

## Why heterogeneity matters

In meta-analysis, the heterogeneity estimate typically affects the weighting of the effect sizes in the primary studies and thereby the estimate of the overall effect size (see Methods for greater detail). Moreover, heterogeneity is of considerable interest in itself because of its practical and epistemic implications. On a practical level, large (unaccounted) heterogeneity means that the effectiveness of an intervention varies strongly and unpredictably across contexts, which is obviously undesirable. Large heterogeneity also reduces the statistical power of studies and should therefore be factored into sample size planning [13, 14]. Finally, heterogeneity also reflects on the state of knowledge in a particular research area. Explained heterogeneity represents progress in knowledge. Often however understanding of heterogeneity remains poor, and in this case large heterogeneity points to a fundamental lack in the understanding of the subject matter [15]. For these reasons, the degree of heterogeneity is of interest in itself, and consequently its correct estimation is important.

## Bias in the set of primary studies

In the absence of pre-registration, effect sizes in published primary studies tend to inflate the underlying population effect sizes [16–19]. Publication bias as well as flexibility in data collection and analysis are driving forces behind this, and we address them in turn. Publication bias arises when studies with statistically non-significant results have a reduced chance of being published [20]. This leads to inflated effect sizes in published primary studies. In unbiased samples, over- and underestimation of the population effect size cancel each other out. But the overestimating samples (e.g., those that find a particularly large difference between the means of experimental and control group) tend to result in lower *p*-values than the underestimating samples. Consequently, under publication bias more overestimating than underestimating samples pass through the publication bottleneck.

Publication bias provides an incentive for researchers to produce statistically significant findings. Given that larger sample effects tend to produce lower *p*-values, researchers might collect and analyse data in ways that lead to systematic overestimation of the population effect

in their sample and thereby push their *p*-value under the threshold for statistical significance [21]. Such practices have become known as *p-hacking* [22]. Their unifying characteristic is that multiple analyses are run by the researcher but only the one that results in the smallest *p*-value is reported. Following earlier work [10], we focus on four practices of *p*-hacking that appear to be widely used in psychology [23]. i) *Optional dependent variables* means that multiple related outcome variables are analysed in a study. ii) *Optional stopping* means that researchers regularly peek at their results and stop data collection when they reach a statistically significant finding (or run out of steam). iii) *Optional moderators* means that the data are sliced in various ways (e.g., all participants; females only; males only). iv) *Optional outlier removal* means that analyses are performed both on all data and on data cleared from outliers.

## Effects on meta-analyses

Publication bias and *p*-hacking lead to inflated effect size estimates in the published literature and in meta-analyses that rely on it [10–12], and this problem is not readily solved by the inclusion of unpublished results [24]. Less is known about their effects on heterogeneity estimates. We are aware of only three studies into the effect of publication bias, and studies on *p*-hacking seem to be missing entirely. Using mathematical reasoning, two studies [25, 26] demonstrated that publication bias might lead to under- or overestimation of heterogeneity. However, this modelling assumed that the censoring of studies is contingent on their effect sizes instead of their statistical significance, which might be unrealistic [27]. A third study using both mathematical reasoning and computer simulations [28] considered the effect of publication bias (which was contingent on statistical significance), while also manipulating the level of true heterogeneity, the magnitude of true effects, and how much studies differed in their sample sizes. A complex picture emerged, but underestimation of heterogeneity was more prevalent than overestimation. The latter was mostly restricted to small effect sizes and tended to increase with the strength of publication bias.

Here, we expand on previous work in four ways. Our first aim is to investigate multiple heterogeneity estimators and compare their performance in a biased world. Our second aim is to investigate publication bias from a new angle. Previous analyses based publication bias on 1-tailed testing, whereby only positive results (i.e., those that point in the desired direction) can escape censoring [25, 28]. In applied research (e.g., medical trials), the valence of effect direction is often unequivocal (e.g., when the treatment reduces or increases mortality). In this case, the allure of positive findings is clear and 1-tailed publication bias appears indisputable. But in some areas of basic research, 2-tailed publication bias might be plausible because findings that go against the grain of received opinion can have particular appeal [29]. Our third aim is to consider the effects not only of publication bias but also of *p*-hacking. Our fourth and last aim is to investigate if biases in sets of primary studies affect estimates of effect size and heterogeneity to a similar degree or if one is prone to stronger distortions than the other.

## Methods

### Random effects model

In meta-analysis, random effects models, which take into account heterogeneity in the effect sizes underlying pertinent primary studies, are often most appropriate [4, 30, 31]. The random effects model describes $\theta_i$, the true effect size in the *i*th study, as

$$\theta_i = \theta + \delta_i \tag{1}$$

whereby $\theta$ is the average true effect size and $\delta_i$ reflects its heterogeneity. The empirically

observed effect size in the $i$th study serves as $\hat{\theta}_i$, which is the estimate for $\theta_i$, and is modelled as

$$\hat{\theta}_i = \theta_i + \varepsilon_i \tag{2}$$

whereby $\varepsilon_i$ is the within study error. $\delta_i$ and $\varepsilon_i$ are typically presumed to be normally distributed with means of zero and variance $\tau^2$ and $\sigma_i^2$, respectively. The average true effect size can then be estimated as

$$\hat{\theta} = \sum_{i=1}^{k} w_i \hat{\theta}_i / \sum_{i=1}^{k} w_i \tag{3}$$

with $k$ being the number of studies in the meta-analysis and $w_i$ their weights. Ideally, weights $w_i = 1/(\sigma_i^2 + \tau^2)$ would be used. However, $\sigma_i^2$ and $\tau^2$ are both unknown and need to be estimated from data.

## Heterogeneity estimators

Numerous methods have been proposed to derive the estimated heterogeneity variance ($\hat{\tau}^2$). We considered five heterogeneity estimators in our simulation, which have either been frequently used or were positively evaluated in relevant reviews [7, 32]: DerSimonian-Laird (DL) [33], Hunter-Schmidt (HS) [34], maximum likelihood (ML) [35], Paule-Mandel (PM) [36], and restricted maximum likelihood (REML) [37].

DL and PM are methods-of-moments estimators and have the general form of

$$\hat{\tau}^2 = \frac{\sum_{i=1}^{k} w_i (\hat{\theta}_i - \hat{\theta})^2 - \sum_{i=1}^{k} w_i \hat{\sigma}_i^2 + \frac{\sum_{i=1}^{k} w_i^2 \hat{\sigma}_i^2}{\sum_{i=1}^{k} w_i}}{\sum_{i=1}^{k} w_i - \frac{\sum_{i=1}^{k} w_i^2}{\sum_{i=1}^{k} w_i}} \tag{4}$$

whereby $\hat{\theta} = \sum_{i=1}^{k} w_i \hat{\theta}_i / \sum_{i=1}^{k} w_i$. DL uses fixed-effects weights, $w_i = 1/\hat{\sigma}_i^2$. In contrast, PM uses random-effects weights $w_i = 1/(\hat{\sigma}_i^2 + \hat{\tau}^2)$, which are determined through an iterative process, which always converges. Using fixed-effects weights $w_i = 1/\hat{\sigma}_i^2$, HS estimates the heterogeneity variance as

$$\hat{\tau}^2 = \frac{\sum_{i=1}^{k} w_i (\hat{\theta}_i - \hat{\theta})^2 - k}{\sum_{i=1}^{k} w_i} \tag{5}$$

ML and REML both employ random-effects weights $w_i = 1/(\hat{\sigma}_i^2 + \hat{\tau}^2)$. ML takes the form

$$\hat{\tau}^2 = \frac{\sum_{i=1}^{k} w_i^2 \left( (\hat{\theta}_i - \hat{\theta})^2 + \hat{\sigma}_i^2 \right)}{\sum_{i=1}^{k} w_i^2} \tag{6}$$

whereas REML uses

$$\hat{\tau}^2 = \frac{\sum_{i=1}^{k} w_i^2 \left( (\hat{\theta}_i - \hat{\theta})^2 + \hat{\sigma}_i^2 \right)}{\sum_{i=1}^{k} w_i^2} + \frac{1}{\sum_{i=1}^{k} w_i} \tag{7}$$

ML and REML both use iterative cycles to jointly estimate $\hat{\theta}_i^2$ and $\hat{\tau}^2$. Occasionally, these fail to converge on a solution. All estimators set any negative values for $\hat{\tau}^2$ to zero.

## Simulation

Simulations were carried out in R (version 4.0.3). *Metafor* [38] version 2.4–0 was used to run meta-analyses on the simulated studies. The annotated R code is available in the supplement.

## Simulation methods

In the simulations, multiple independent studies were run and submitted for publication (potentially biased by $p$-hacking), published (or not), and (if published) summarized in a meta-analysis. Between-subjects experiments with two groups were simulated. The outcome variable was continuous, and the standardized mean difference (SMD) served as effect size index. Meta-analyses on continuous outcomes are frequent in psychology [1]. We aggregated observed sample sizes from a representative set of 150 psychological meta-analyses [15] into a single distribution. Sample sizes $N_i$ for simulated studies were randomly sampled from this distribution and equally split between groups 1 and 2. Median $N_i$ (for both groups combined) was 100, with an interquartile range of 176. If average sample size differed considerably across the 150 meta-analyses in our set, our approach might result in unrealistic combinations of very large and very small samples in simulated meta-analyses, which in turn might distort our results [28]. However, an ANOVA (bias corrected accelerated bootstrap with 1,000 samples) revealed little variation of average sample size across these 150 meta-analyses ($\eta_p^2 = 0.020$, $F$ $(149, 7077) = 0.97$, $p = .595$).

## Describing heterogeneity

We use $\tau$ to describe heterogeneity. In the present context, it has a number of advantages. In contrast to $\tau^2$, $\tau$ has an intuitive interpretation in that it reflects the standard deviation of the true effect size. Moreover, $\tau$ is in the same SMD unit as the simulation's effect size estimates. This facilitates our fourth aim, to compare the effects of biased sets of primary studies on estimates of effect size and estimates of heterogeneity. Imagine that a given level of publication bias and $p$-hacking led to a bias of 0.1 in the overall effect size estimate $\hat{\theta}$ and a bias of 0.1 in the heterogeneity estimate $\hat{\tau}$. In this case it would be sensible to conclude that effect size estimates and heterogeneity estimates were affected to the same extent (although the same degree of bias might be seen as more consequential for effect size estimates than for heterogeneity estimates).

To describe bias in heterogeneity estimates we found verbal labels helpful, although a degree of arbitrariness is inevitable. In a recent survey of heterogeneity in psychology meta-analyses, average $T$ (the empirical estimate of heterogeneity in SMD units) was 0.33 [15]. In light of this, labels of small/medium/large for (unsigned) bias in $T$ of 0.05/0.10/0.20 struck us as sensible and we will use them in this way throughout.

## Factors manipulated

To address our first aim, we compared the performance of the five heterogeneity estimators described above. Six factors were manipulated in our simulations (see Table 1). Addressing

**Table 1. Simulation parameters.**

| Experimental Factors | Abbreviation | Levels |
|---|:---:|:---:|
| $P$-hacking | $p$-hack | None, medium, high |
| Type of publication bias | TAIL | 1-tailed, 2-tailed |
| Strength of publication bias[a] | PB | 0%, 40%, 80% |
| True heterogeneity | $\tau$ | 0, 0.11, 0.22, 0.33, 0.44 |
| True average effect size | $\theta$ | 0, 0.2, 0.5, 0.8 |
| Number of studies per meta-analysis | $k$ | 9, 18, 36, 72 |

[a]Indicated as the proportion of statistically non-significant studies that remain unpublished. For 1-tailed publication bias, all negative findings are censored, independent of the strength of publication bias.

our second aim, the first factor concerned the type and prevalence of *p*-hacking applied to the experiments. We considered the four types of *p*-hacking [10] described above. i) Optional dependent variables: Researchers in the simulated experiment used two dependent variables, which were correlated ρ = .8 at the population level. ii) Optional stopping: After reaching their starting $N_i$, researchers regularly peeked at their results. They kept adding 10% of the starting $N_i$ until they either obtained a statistically significant finding or hit maximum $N_i$ (arbitrarily set at five times the starting $N_i$ or 200, whichever is lower). iii) Optional moderator: The sex of all participants in the experiment was decided at random (*p* = .5). Researchers analyzed results for females only, males only, and the whole sample. iv) Optional outlier removal: Researchers run separate analyses on all data, and on data with outliers (unsigned z ≥ 2) removed.

We then used these four *p*-hacking strategies to simulate three research environments [10]: no, medium, and high *p*-hacking. In the no *p*-hacking environment, no *p*-hacking was used. Consequently, each experiment leads to only one result, which would be published (unless censored by publication bias). In the medium *p*-hacking environment, 30% of researchers did not engage in *p*-hacking, 50% of researchers used both optional dependent variables and optional stopping, and 20% of researchers used all four *p*-hacking strategies. For the high *p*-hacking environment, these percentages were 10%, 40%, and 50%, respectively. Multiple *p*-hacking strategies were fully crossed. Thus, a researcher who engaged in all four would first study starting $N_i$ participants and perform analyses on both dependent variables, with and without outliers, on all participants and on females and males separately. If none of these 12 analyses returned a statistically significant result (*p* < .05), 10% more participants were studied, and the same analyses carried out again. This cycle ended when either statistical significance or maximum $N_i$ was reached. If multiple analyses resulted in statistically significant results at this point, only the one with the smallest *p*-value was submitted for publication.

Addressing our third aim, the second factor implemented type of publication bias as either 1- or 2-tailed. Under 1-tailed publication bias, statistically significant results (2-tailed testing) in the expected direction were always published; all other results were censored to a degree that was defined by the strength of publication bias. If *p*-hacking required selection between multiple analyses, this was contingent on a modified *p*-value, which equaled *p* for results in the expected direction. For results in the opposite direction, the modified *p*-value was computed as 1 + (1-*p*). Obviously, the modified *p*-value cannot be interpreted as a probability, but it appropriately penalizes results in the wrong direction with, ceteris paribus, stronger effects carrying greater penalties. With 2-tailed publication bias, statistically significant results were published regardless of sign, and all other results were censored to a degree that was defined by the strength of publication bias. Strength of publication bias was the third factor, implemented with levels 0%/40%/80% of non-significant results being censored. Degree of true heterogeneity was implemented with levels τ = 0.00 to 0.44 in steps of 0.11. The three highest levels represent average heterogeneity ±1SD observed in a survey of psychological meta-analyses [15]. We decided against inclusion of higher levels in our simulation because the usefulness of the meta-analytic model becomes questionable in the face of very high heterogeneity [30]. To what extent multiple close replications (in which an original study is replicated as faithfully as possible across many labs and each lab's results are treated as a separate study) show heterogeneity has become an important issue in psychology [39]. We therefore included the additional level of τ = 0.11, which comes close to average observed heterogeneity in multiple close replications [15]. To facilitate comparisons with meta-analyses that express heterogeneity in the $I^2$ metric (i.e., the proportion of between-study variance estimated to be due to heterogeneity), we computed mean $I^2$ levels across simulations without publication bias and questionable research practices. The simulation's five heterogeneity levels translated into $I^2$ values of 6.7%, 31.7%, 62.4%, 78.4%, and 86.2% (with respective medians of 0.0%, 30.4%, 65.8%, 81.3%, and 88.3%).

In Monte Carlo simulations of bias-free meta-analyses, the strength of the true effect can typically be disregarded as inconsequential [6, 37, 40]. In a biased world, however, $\theta$ proves important [10, 28]. We therefore implemented $\theta$ and used levels 0.0/0.2/0.5/0.8 because the latter three are often considered as benchmarks for small/medium/large effects in psychology [41]. Finally, the number of studies feeding into each meta-analysis was implemented with $k = 9/18/36/72$; an average of $k = 37$ in psychology meta-analyses motivated these choices [15].

A summary of simulation factors is provided in Table 1. All six manipulated factors were fully crossed, resulting in 1,440 unique factor combinations. Following [10], 1,000 meta-analyses were run for each (due to the intense computational demands in the conditions that involved questionable research practices, a higher number did not prove feasible). Occasional trials in which ML or REML failed to converge on a solution were replaced until 1,000 meta-analyses were completed. For each cell of the design, the simulation computed the standard deviation across the 1,000 heterogeneity estimates. Following [42], we divided this by $\sqrt{1000}$ to estimate the Monte Carlo error, i.e., the standard error for the heterogeneity estimate in each cell. The mean was 0.0023, its maximum 0.0068, which strikes us as sufficiently low. The annotated R code, which provides further technical details, is available at https://osf.io/qga8v/.

## Results

Throughout our results, we will refer to estimates of overall effect size as $d$ and to estimates of heterogeneity as $T$. Data files are available here: https://osf.io/qga8v/.

P-hacking increased mean $N_i$ in simulated meta-analyses from 123 (no p-hacking) to 141 (medium p-hacking) and 129 (high p-hacking).

### Estimation of effect size and heterogeneity in the absence of bias

To evaluate estimation performance in the absence of bias, analyses in this section are restricted to simulation conditions without p-hacking and without publication bias. Across analyses and in line with previous findings, level of effect size proved of little consequence [6]. Consequently, figures do not differentiate results by effect size.

Estimates of $\theta$ proved virtually unbiased (see S1 Fig), which is in line with previous simulations [6, 40, 43, 44]. Coverage probability (i.e., the percentage of confidence intervals that included θ) was too low for lower (but not for absent) heterogeneity, particularly in conjunction with small $k$ (see S2 Fig). As can also be seen, the HS and ML estimator suffered worst from these problems. Consequently, type-1 errors were inflated under the same circumstances (see Fig 1), again particularly for the HS and ML estimator. This contrasts with previous findings conducted with simulations with similar parameters to ours [40], with the notable exception of markedly lower variability in $N_i$ in their study. For a variety of heterogeneity estimators, including for HS and ML, they found excellent coverage for confidence intervals based on $t$-distributions.

Unlike estimates of $\theta$, heterogeneity estimates proved somewhat biased. True levels of $\tau = 0$ were overestimated (which is not surprising, given that heterogeneity estimates are $\geq 0$); for all other heterogeneity levels, $\tau$ estimates proved too low, especially when $k$ was low (see Fig 2). Again, these problems were particularly strong for the HS and ML estimator. Bias for the DL, PM, and REML estimator rarely exceeded 0.02 and therefore appears negligible, particularly in light of average heterogeneity ($\tau = 0.33$) in a survey of meta-analyses on continuous outcomes [15].

Low bias is only one desirable property for heterogeneity estimators. In addition, they should be little affected by sampling fluctuation, i.e., under the same circumstances the variance in their estimates should be low. The root mean square error for heterogeneity estimates

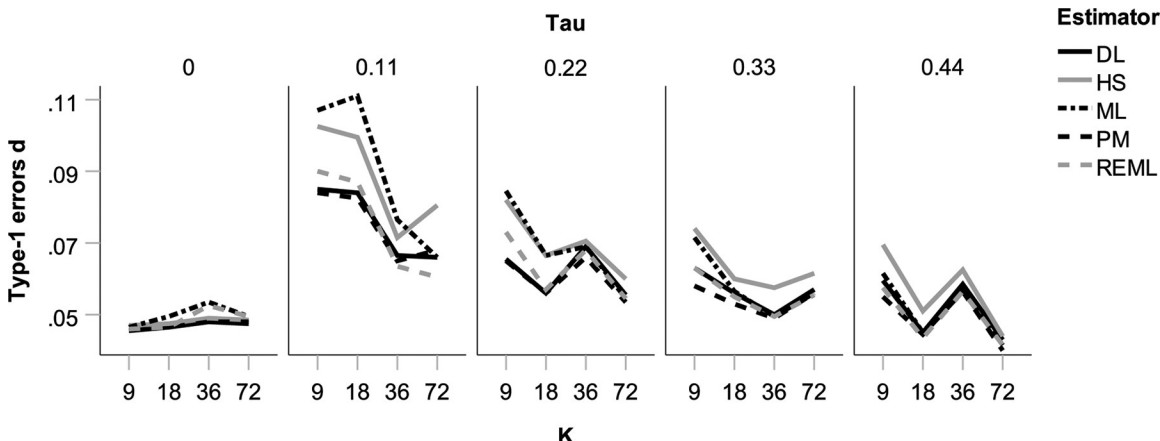

**Fig 1. Proportion of type-1 errors for overall effect size estimate d in the absence of publication bias and *p*-hacking for five heterogeneity estimators as a function of true heterogeneity (τ) and number of studies per meta-analysis (k), with α-level = 0.05.**

($T_{\text{rmse}}$) combines both bias and variance. By this measure, the ML and REML estimators performed consistently well (see Fig 3). The DL estimator, although showing little bias (see Fig 2), lost ground through relatively large variance, particularly for larger *k*; conversely, the ML estimator, although showing considerable bias (see Fig 2), looked somewhat better on $T_{\text{rmse}}$ because of its low variance (see S3 Fig). Our findings on bias and RMSE are broadly in line with those of previous simulations [32]. (A notable exception is a previous study [37] that found ML and HS to be comparable on both criteria, whereas HS performed clearly worse in our simulation, particularly for larger heterogeneity, as shown in Figs 2 and 3. This discrepancy might be partly down to the fact that the previous study implemented somewhat weaker heterogeneity (τ ≤ 0.31) than our simulation (τ ≤ 0.44) and did not truncate negative heterogeneity estimates to zero.) Finally, coverage of confidence intervals around *T* proved excellent across all conditions (see S4 Fig).

To summarize, in the absence of biases we found two problems in estimations: First, considerable type-1 error inflation occurred in tests of the overall effect size when true heterogeneity was low. This occurred even though these tests implemented the Knapp-Hartung adjustment [45]. Second, the HS estimator (and, to a lesser extent, the ML estimator) led to considerable underestimation of heterogeneity at the highest level of true heterogeneity in

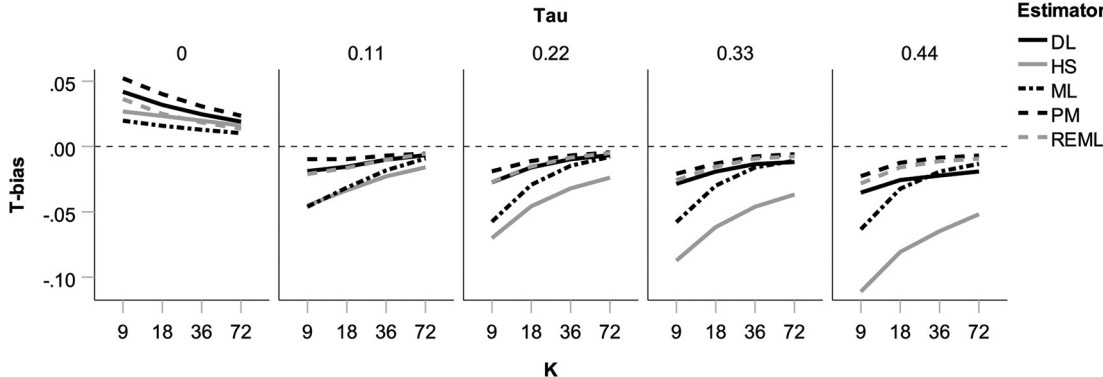

**Fig 2.** Bias in heterogeneity estimates (Tbias) in the absence of publication bias and p-hacking for five heterogeneity estimators as a function of true heterogeneity (τ) and number of studies in the meta-analysis (k).

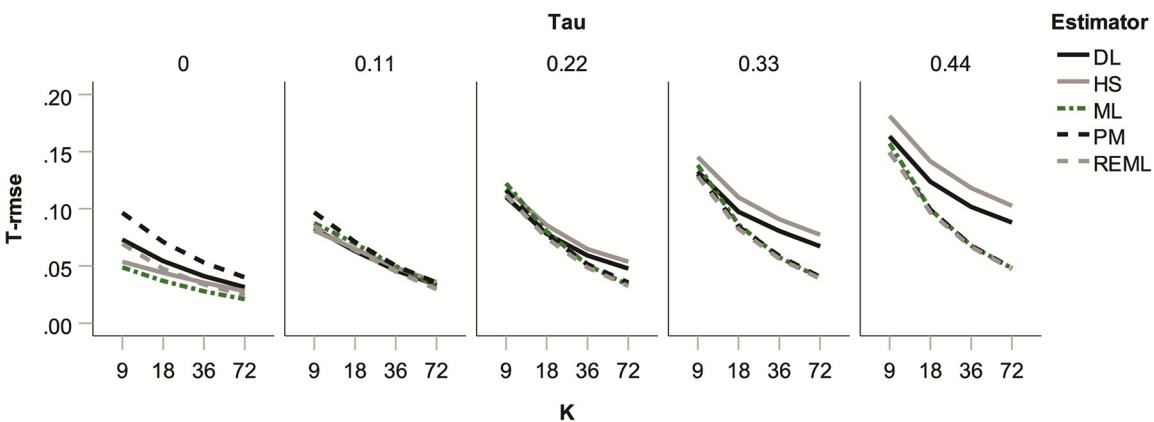

**Fig 3. Root mean square error for heterogeneity estimates ($T_{\text{RMSE}}$) in the absence of publication bias and *p*-hacking for five heterogeneity estimators as a function of true heterogeneity ($\tau$) and number of studies in the meta-analysis ($k$).**

connection with low *k*. Overall, the REML estimator performed particularly well due to a combination of low bias and low variance. PM demonstrated the same strengths, unless heterogeneity was absent, which will be an unrealistic assumption in most contexts. Our simulations thus support pervious positive evaluations of the REML and PM estimators in the absence of bias [6, 32].

## Estimation of heterogeneity in the presence of bias

In this section we look at heterogeneity estimates across all levels of our simulation and start with effects on $T_{\text{bias}}$ (i.e., $T-\tau$). Given the complexity of our simulation, understanding which factors or factor combinations matter poses a challenge. To address this problem we ran, for each heterogeneity estimator, a six-factorial between-subjects ANOVA on $T_{\text{bias}}$ and used effect sum of squares to understand which factors and interactions proved most influential. In this and subsequent ANOVAs, main effects, 2-way-interactions, and 3-way interactions together accounted for upwards of 98% of variance for each estimator. Here and in subsequent analyses, we can therefore exclude an important role for 4-way and higher interactions, and consequently we do not comment on them. Table 2 identifies the most important effects. However, before we consider them in detail it is of interest to identify which heterogeneity estimators were least and most affected by our manipulations. (We will refer to this characteristic as an estimator's "inertia" vs. "volatility".) Ideally, any heterogeneity estimator should be rather inert. If its bias is low, inertia instils confidence that low bias will also prevail under the specific (but largely unknown) circumstances for the meta-analysis at hand. (Note that a meta-analyst only knows $N_{\text{i}}$ and *k* for sure. The prevalence of *p*-hacking, the true heterogeneity between studies, etc. remain unknown.) If the estimator's bias is large, inertia implies that it could be confidently corrected. The ANOVA's corrected total sums of squares directly reflect estimators' volatility. As can be seen from Table 2, the DL estimator proved most inert, whereas PM was (by a considerable margin) most volatile.

Regarding main effects on $T_{\text{bias}}$, strength of publication bias and *k* proved largely inconsequential (see Table 2). In decreasing order of importance, tau, effect size, type of publication bias, and *p*-hacking prevalence proved relevant. Their effects are summarized in the panels of Fig 4. As can be seen, all five heterogeneity estimators were affected in similar ways. Overall, $T_{\text{bias}}$ was driven upwards by lower levels of true heterogeneity, the absence of a true effect, 2-tailed publication bias, and higher levels of *p*-hacking. In general, the PM estimator

**Table 2. The relative importance of design factors for $T_{bias}$.** Selected sum of squares from six-factorial ANOVA for five heterogeneity estimators.

|  | DL | HS | ML | PM | REML | M |
|---|---|---|---|---|---|---|
| **P-hack** | 0.25 | 0.22 | 0.36 | 1.03 | 0.40 | 0.45 |
| **TAIL** | 0.48 | 0.37 | 1.03 | 1.65 | 1.11 | 0.93 |
| **θ** | 0.40 | 0.36 | 1.06 | 2.13 | 1.13 | 1.01 |
| **τ** | 1.88 | 3.39 | 1.13 | 1.89 | 1.04 | 1.87 |
| **k** | 0.01 | 0.25 | 0.27 | 0.01 | 0.02 | 0.11 |
| **PB** | 0.01 | 0.00 | 0.06 | 0.07 | 0.06 | 0.04 |
| *Sum main effects* | *3.03* | *4.59* | *3.91* | *6.78* | *3.76* | *4.41* |
| **θ × PB** | 0.20 | 0.14 | 0.42 | 0.51 | 0.46 | 0.35 |
| **P-hack × θ** | 0.02 | 0.02 | 0.10 | 0.22 | 0.10 | 0.09 |
| **TAIL × θ** | 0.27 | 0.21 | 0.72 | 0.97 | 0.77 | 0.59 |
| **P-hack × TAIL** | 0.10 | 0.08 | 0.30 | 0.53 | 0.33 | 0.27 |
| **P-hack × τ** | 0.15 | 0.11 | 0.10 | 0.24 | 0.12 | 0.14 |
| . . . |  |  |  |  |  |  |
| *Sum 2-way interactions* | *1.02* | *0.82* | *2.10* | *2.88* | *2.22* | *1.82* |
| **P-hack × TAIL × θ** | 0.07 | 0.05 | 0.23 | 0.32 | 0.25 | 0.18 |
| . . . |  |  |  |  |  |  |
| *Sum 3-way interactions* | *0.24* | *0.16* | *0.54* | *0.63* | *0.55* | *0.39* |
| *Error* | *0.03* | *0.02* | *0.07* | *0.05* | *0.07* | *0.05* |
| *Corrected total* | *4.29* | *5.62* | *6.60* | *10.34* | *6.61* | *6.69* |

The table shows all main effects and all 2-way and 3-way interactions for which sum of squares ≥ 0.20 for at least one estimator. The rightmost column shows the mean across the five estimators.

produced the highest heterogeneity estimates and HS the lowest, with DL, ML, and REML in between. This meant that the largest positive levels of $T_{bias}$ were observed for PM and HS. The PM estimator showed small to moderate positive $T_{bias}$ for absent or low heterogeneity, for an absent or small true effect, for 2-tailed publication bias and for moderate and high $p$-hacking. The HS estimator showed small negative $T_{bias}$ for higher levels of heterogeneity, for 1-tailed publication bias, and in the absence of $p$-hacking. DL, ML, and REML showed the largest (small, positive) $T_{bias}$ in the absence true heterogeneity and in the absence of a true effect.

Regarding 2-way interactions, effect size × type of publication bias and effect size × strength of publication bias were most relevant (see Table 2) and are shown in Figs 5 and 6. In the absence of a true effect, 2-tailed publication bias increased heterogeneity estimates particularly strongly and positive $T_{bias}$ emerged (small for HS and DL; moderate for ML and REML; and

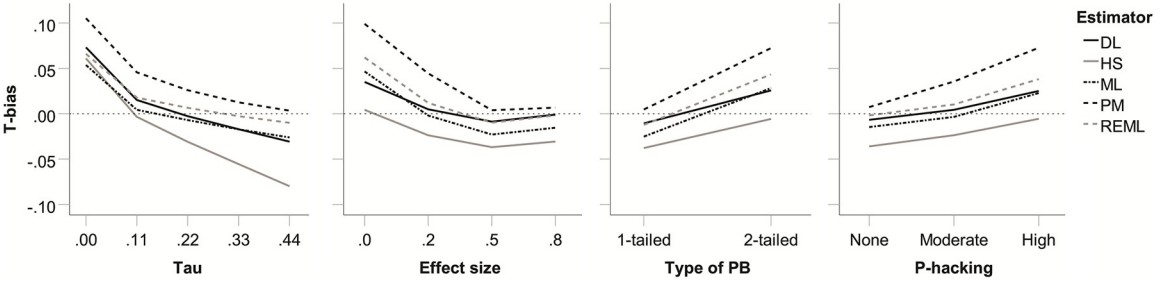

**Fig 4. Bias in heterogeneity estimates ($T_{bias}$) for five heterogeneity estimators as a function of true heterogeneity ($\tau$), true average effect size ($\theta$), type of publication bias, and $p$-hacking environment, respectively.**

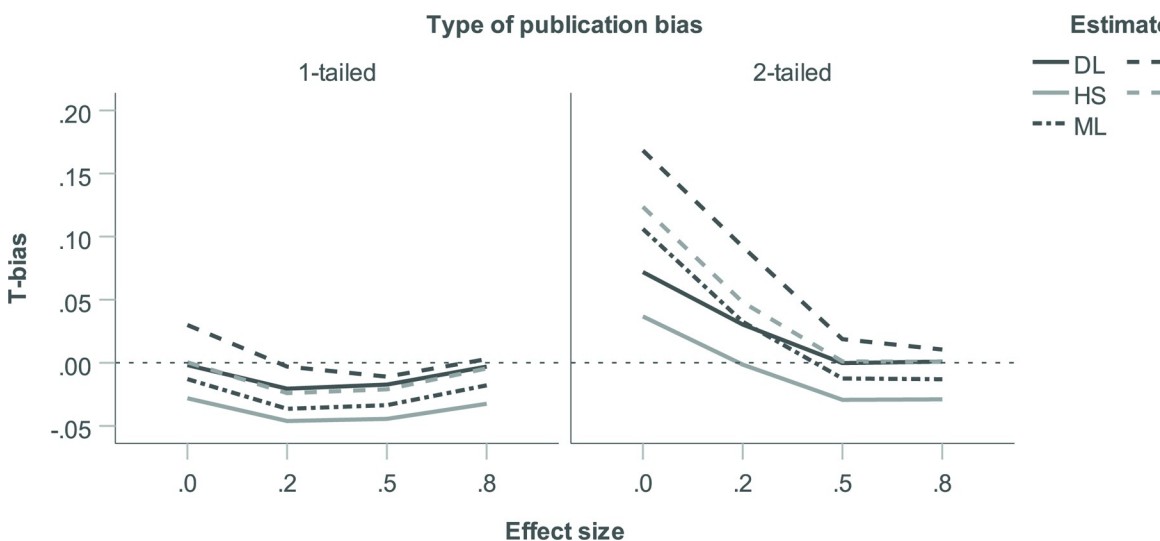

**Fig 5. Bias in heterogeneity estimates ($T_{bias}$) for five heterogeneity estimators: Two-way interaction of true average effect size ($\theta$) with type of publication bias.**

moderate-to-large for PM; see Fig 5). This arises because only under 2-tailed publication bias do $p$-hacking and publication bias have the potential to push published effect sizes either above or below zero, thus maximising their variance. Similarly, the absence of a true effect also boosted the positive $T_{bias}$ created by strong publication bias (see Fig 6). At 80% publication bias and in the absence of a true effect, positive $T_{bias}$ was moderate-to-large for the PM estimator, moderate for ML and REML, small for DL, and least pronounced for HS. This reflects that strong publication bias maximises the variance in published effect sizes at $\theta = 0$ because (exaggerated) published effect sizes are equally likely to be above or below zero. Other effects on $T_{bias}$ proved moderate in size and are immaterial to our discussion, but for illustration the largest 3-way interaction is shown in S5 Fig.

Because our simulation considered for the first time $p$-hacking in addition to publication bias, we compared their effects in greater detail. In the ANOVA (Table 2), the 2-way

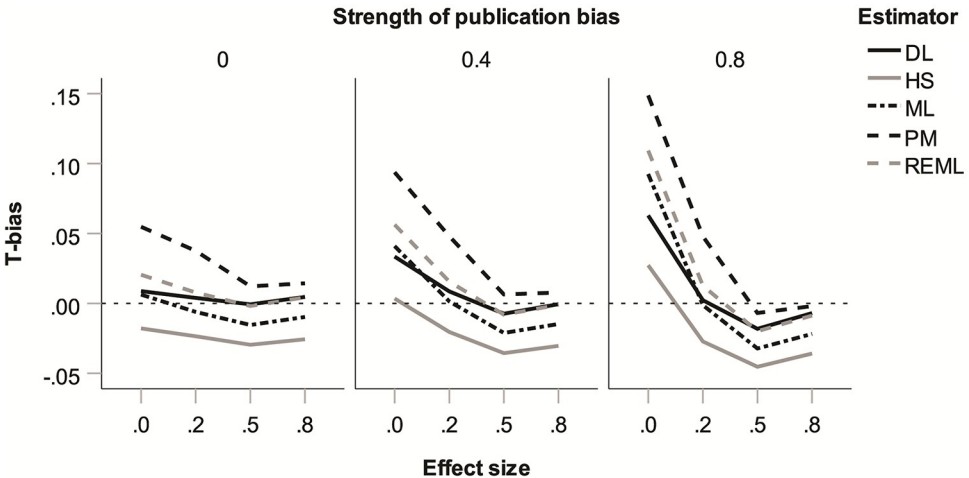

**Fig 6. Bias in heterogeneity estimates ($T_{bias}$) for five heterogeneity estimators: Two-way interaction of true average effect size ($\theta$) with strength of publication bias.**

**Table 3. The relative importance of design factors for $T_{rmse}$.** Selected sum of squares from six-factorial ANOVA for five heterogeneity estimators.

|  | DL | HS | ML | PM | REML | M |
|---|---|---|---|---|---|---|
| *P*-hack | 0.11 | 0.05 | 0.27 | 0.84 | 0.35 | 0.33 |
| TAIL | 0.02 | 0.00 | 0.11 | 0.61 | 0.21 | 0.19 |
| θ | 0.07 | 0.02 | 0.29 | 0.84 | 0.42 | 0.33 |
| τ | 0.71 | 1.25 | 0.43 | 0.28 | 0.33 | 0.60 |
| *k* | 0.63 | 0.53 | 0.91 | 1.13 | 1.00 | 0.84 |
| PB | 0.03 | 0.03 | 0.10 | 0.14 | 0.12 | 0.08 |
| *Sum main effects* | *1.58* | *1.88* | *2.11* | *3.85* | *2.42* | *2.37* |
| *P*-hack × TAIL | 0.03 | 0.02 | 0.15 | 0.26 | 0.16 | 0.13 |
| *P*-hack × θ | 0.04 | 0.02 | 0.24 | 0.55 | 0.32 | 0.23 |
| TAIL × θ | 0.12 | 0.15 | 0.13 | 0.21 | 0.12 | 0.15 |
| τ × θ | 0.03 | 0.02 | 0.15 | 0.26 | 0.16 | 0.13 |
| . . . |  |  |  |  |  |  |
| *Sum 2-way interactions* | *0.46* | *0.52* | *1.03* | *1.82* | *1.07* | *0.98* |
| *P*-hack × TAIL × θ | 0.02 | 0.01 | 0.10 | 0.20 | 0.12 | 0.09 |
| . . . |  |  |  |  |  |  |
| *Sum 3-way interactions* | *0.16* | *0.18* | *0.39* | *0.45* | *0.38* | *0.31* |
| *Error* | *0.03* | *0.03* | *0.06* | *0.05* | *0.06* | *0.05* |
| *Corrected total* | *2.22* | *2.61* | *3.60* | *6.17* | *3.93* | *3.71* |

The table shows all main effects and all 2-way and 3-way interactions for which sum of squares $\geq 0.20$ for at least one estimator.

interaction between *p*-hacking and strength of publication bias proved zero for all five heterogeneity estimators. In other words, the effects of *p*-hacking and publication bias were strictly additive, which is shown in S6 Fig. Nonetheless, their interactions with effect size proved somewhat different in nature (see S7 Fig). For an effect size of zero, both higher levels of *p*-hacking and stronger publication bias strongly increased $T_{bias}$. For larger effect sizes, a similar (although slightly weaker) effect emerged for *p*-hacking (see upper panel), but a reversal of this effect was observed publication bias; i.e., for larger effect sizes, an increase in publication bias now led to a (small) *decrease* in $T_{bias}$ (see lower panel).

Unlike previous work [28], we implemented *p*-hacking in addition to publication bias and described heterogeneity via τ instead of $I^2$. These differences notwithstanding, our simulation confirmed their finding that (under 1-tailed publication bias), overestimation of heterogeneity occurs under fewer simulation conditions than underestimation, and the latter is particularly strong for small θ and strong publication bias (see S8 Fig, which is restricted to 1-tailed publication bias).

Next, we look at $T_{rmse}$ to also capture estimators' variance in addition to their bias. Again, we used sum of squares from six-factorial between-subjects ANOVA on $T_{rmse}$ for guidance (Table 3). And as previously, we used ANOVA's corrected total sum of squares to judge estimators' inertia. Paralleling inertia for $T_{bias}$, the DL estimator proved again most inert, whereas PM was again considerably more volatile than any other estimator.

Regarding main effects on $T_{rmse}$, both strength and type of publication bias proved largely inconsequential. In decreasing order of importance, *k*, τ, θ, and *p*-hacking prevalence proved particularly relevant. Again, the main effects affected all five heterogeneity estimators in similar ways, but PM fared generally poorly in comparison to the others. Not surprisingly, $T_{rmse}$ was decreased by increasing *k*, but also by low (but not absent) heterogeneity, by larger effect sizes, and by less prevalence of *p*-hacking (see Fig 7). Across levels of *k*, the performance of DL,

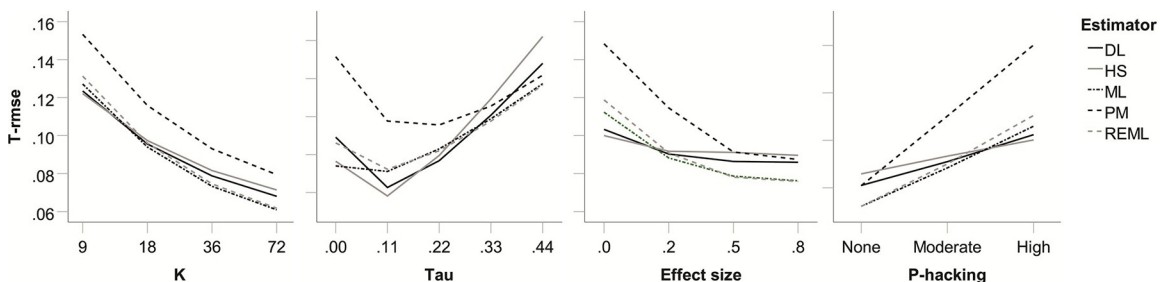

**Fig 7. Root mean square error for heterogeneity estimates ($T_{RMSE}$) for five heterogeneity estimators as a function of number of studies in the meta-analysis ($k$), true heterogeneity ($\tau$) true average effect size ($\theta$), and $p$-hacking environment, respectively.**

HS, ML, and REML proved very similar to each other's. Otherwise, DL and HS as well as ML and REML tended to show similar $T_{rmse}$. DL/HS proved somewhat better for $\tau = 0.11$ and in the absence of a true effect; ML/REML proved somewhat better for $\tau = 0.44$ and for medium and large effects.

The strongest 2-way interaction (of effect size with $p$-hacking) did not add much to the comparison of estimators over and above the main effects just discussed (see S9 Fig).

## Estimation of effect size in the presence of bias

In this section we return to effect size estimates, but this time across all simulation conditions. We focus on $d_{bias}$ (i.e., $d-\theta$, whereby $d$ is the unbiased estimate of Cohen's $d$, see [46]), which was hardly affected by the type of heterogeneity estimator used. For reporting economy, we report results only for (arbitrarily chosen) DL.

**Table 4. The relative importance of design factors for $d_{bias}$ (selected sum of squares from six-factorial between-subjects ANOVA).** Data are shown for DL but were very similar across all heterogeneity estimators.

| | | |
|---|---|---|
| *P*-hack | | 2.18 |
| **TAIL** | | 0.89 |
| $\theta$ | | 0.28 |
| $\tau$ | | 0.89 |
| *k* | | 0.00 |
| PB | | 1.03 |
| *Sum main effects* | | *5.27* |
| $\theta \times$ PB | | 0.21 |
| TAIL $\times \theta$ | | 1.04 |
| TAIL $\times$ PB | | 0.22 |
| $\tau \times$ PB | | 0.22 |
| TAIL $\times \tau$ | 0.26 | |
| . . . | | |
| *Sum 2-way interactions* | | *2.22* |
| TAIL $\times \theta \times$ PB | | 0.30 |
| . . . | | |
| *Sum 3-way interactions* | | *0.82* |
| *Error* | | *0.08* |
| *Corrected total* | | *8.38* |

The table shows all main effects and all 2-way and 3-way interactions for which sum of squares $\geq 0.20$.

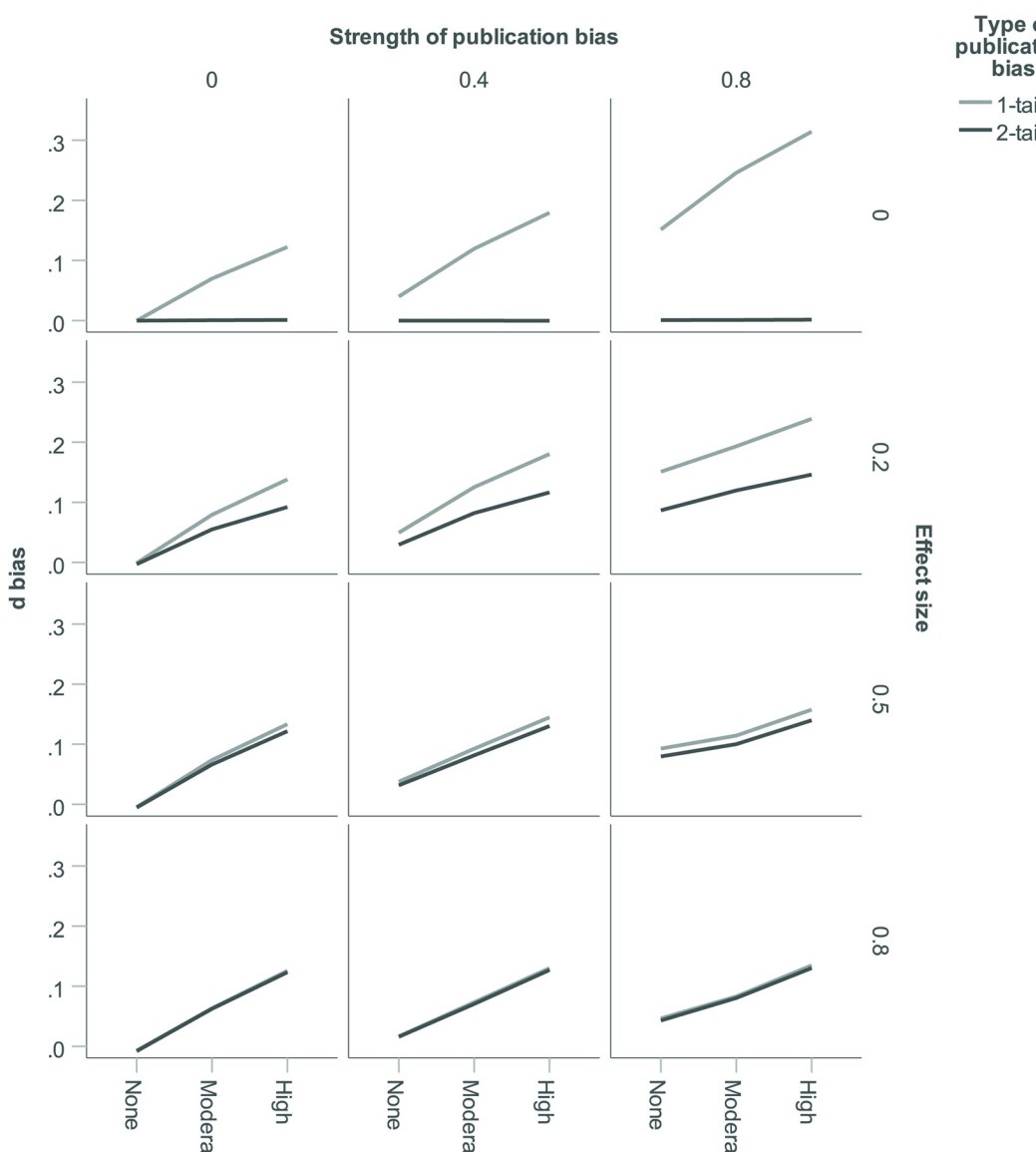

**Fig 8. Bias in effect size estimates ($d_{\text{bias}}$) as a function of $p$-hacking environment, strength of publication bias, true average effect size ($\theta$), and type of publication bias.** Data shown are for the DL estimator but are very similar for other estimators.

As previously, we use sum of squares from six-factorial between-subjects ANOVA on $d_{\text{bias}}$ to understand which simulation factors and interactions were most influential (see Table 4). Effects on $d_{\text{bias}}$ proved somewhat more complex than effects on $T_{\text{bias}}$: Main effects explained only 63% of the variance in $d_{\text{bias}}$ (66% for $T_{\text{bias}}$, averaged across estimators) and 3-way interactions explained 10% (6% for $T_{\text{bias}}$, averaged across estimators). Fig 8 provides an overview over important effects. Obviously, $p$-hacking and publication bias both increased $d_{\text{bias}}$; under the levels selected in our simulation, the former proved more powerful. The combination of both could induce large bias. E.g., for a true effect size of zero, $\theta$ might be estimated to be over 0.3, a substantial effect. As one would expect, $d_{\text{bias}}$ was also stronger under 1-tailed than under 2-tailed publication bias, especially when a true effect was absent or small. Perhaps less intuitively, $d_{\text{bias}}$ also increased with $\tau$ (see S10 Fig).

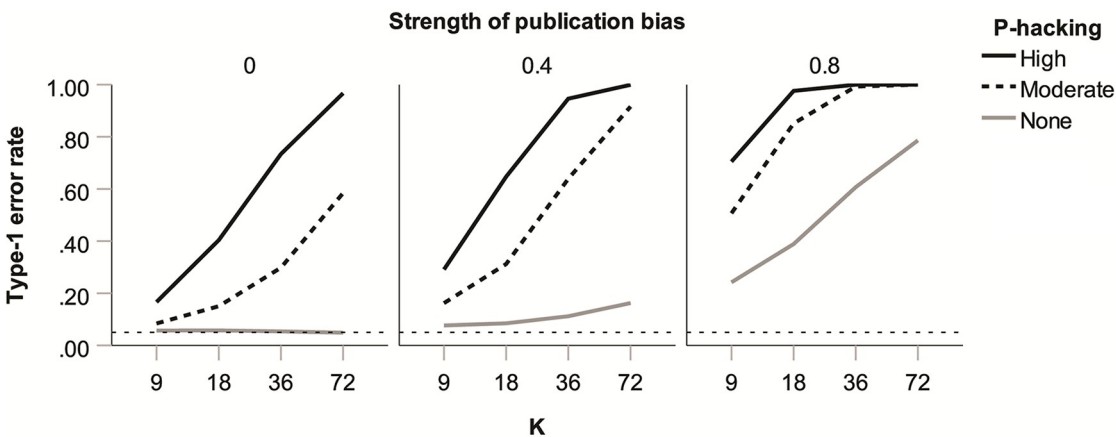

**Fig 9. Type-1 error rates for *d* under 1-tailed publication bias as a function of strength of publication bias, level of *p*-hacking, and number of studies in the meta-analysis (*k*).** Data shown are for conditions with $\theta = 0$ and are based on the DL estimator but are very similar for other estimators.

For overall effect size estimates (*d*), meta-analyses typically report a *p*-value, which is tacitly assumed to provide an appropriate safeguard against type-1 errors. Fig 9 shows type-1 error rates for *d* under 1-tailed publication bias in our simulation. (Under 2-tailed publication bias, type-1 error rates proved very close to the nominal 5%.) As can be seen, type-1 error rates might reach catastrophic levels. Random effects *p*-values for *d* will therefore fail to offer protection against type-1 errors unless publication bias and *p*-hacking can be ruled out.

## Comparison of biases in estimates of effect size and heterogeneity

As we expressed effect size and heterogeneity in the same SMD unit, it is possible to compare the effects of biased research on estimates of effect size and heterogeneity. For this purpose, Fig 10 contrasts unsigned estimation error for *d* and for *T* (i.e., absolute $d_{bias}$ and absolute $T_{bias}$) via boxplots. The upper panel is based on all simulation conditions. The middle panel excludes simulations with $\tau \leq 0.11$ because such low levels of heterogeneity are rarely observed [15]. It also excludes simulations with $\theta = 0$. These might often translate into relatively small effect size estimates, which in turn might render them (and their level of heterogeneity) of little interest to researchers, at least in some areas of psychology. Finally, the lower panel in Fig 10 also excludes 2-tailed publication bias, because this might be unrealistic in many domains. As can be seen, errors in effect size estimation were consistently much larger than errors in heterogeneity estimation. From this perspective, publication bias and *p*-hacking cause much more problems for the estimation of effect size than for the estimation of heterogeneity, especially when the latter relies on the DL, PM, or REML estimator.

## Discussion

One aim of our simulation was to compare the performance of heterogeneity estimators when publication bias and *p*-hacking distort the effect size estimates in primary studies. Confirming previous findings, we found that REML and PM did well when the sets of primary studies were unbiased [6, 32]. However, a different picture emerged once publication bias and *p*-hacking came into play: PM often performed poorly in terms of both bias and RMSE, whereas DL proved least biased while also showing low RMSE. Under conditions that might be particularly realistic and/or relevant in many research contexts (presence of a real effect and considerable

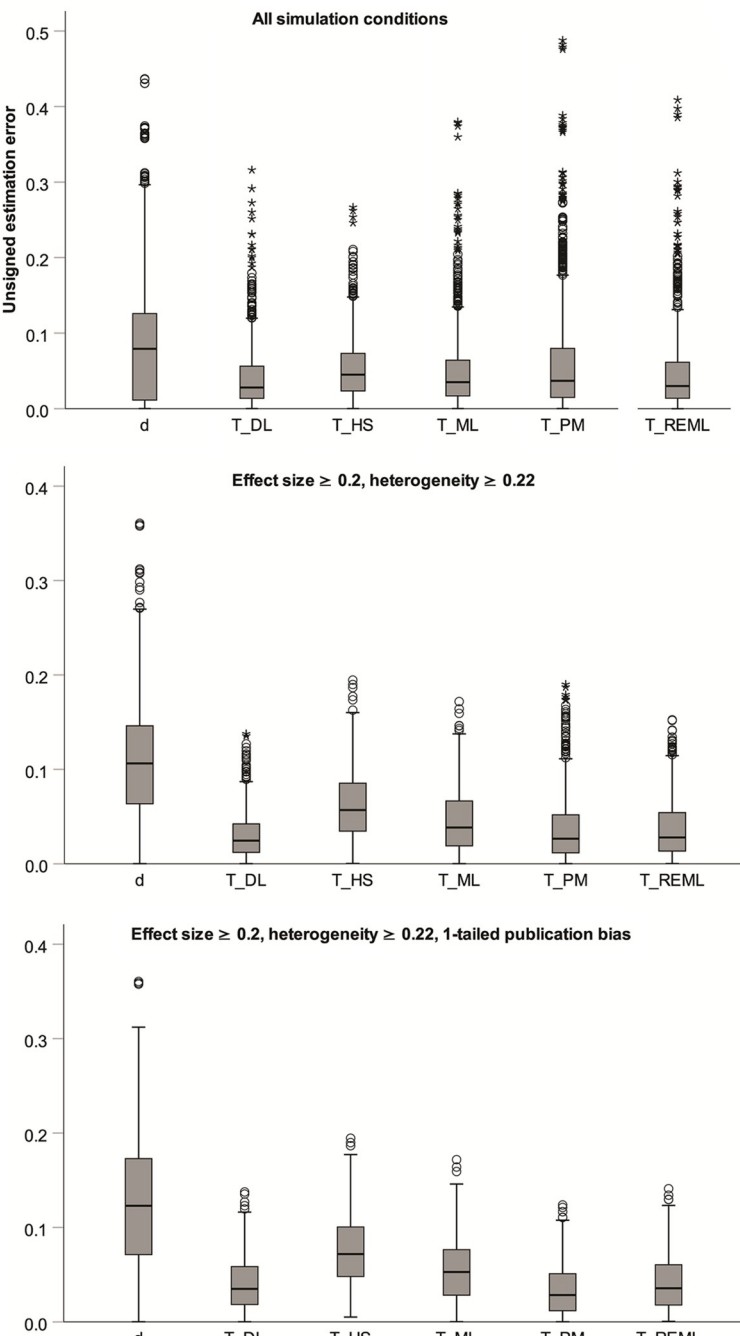

**Fig 10. Comparison of errors in estimation of effect size and heterogeneity.** Estimation errors for *d* are for the DL estimator, but virtually identical for the other estimators.

heterogeneity, any publication bias is 1-tailed), REML showed similarly low levels of bias and lower RMSE.

We also compared the effects of 1-tailed versus 2-tailed publication bias. In line with a previous simulation [28], we found that underestimation of heterogeneity dominated under 1-tailed publication bias. However, overestimation prevailed under 2-tailed publication bias and the PM estimator proved particularly susceptible. Two-tailed publication bias might be

expected only in a limited number of fields in which findings that go against the grain have appeal [29]. Nonetheless we believe this differentiation to be important.

Finally, we thought to compare the effects of biased sets of primary studies on estimates of effect size and heterogeneity. In the bias-free world that underlies most investigations on this subject matter, estimation of heterogeneity proves much more challenging than effect size estimates [32]. However, the presumed absence of biases in sets of primary studies seems unrealistic in many fields [8, 9, 19, 23]. In our simulation, biased sets of primary studies caused much more severe problems for estimates of effect size than for estimates of heterogeneity. Therefore, future investigations into meta-analytic parameter estimations should prioritise how to deal with biases in effect size estimates [e.g., 12, 22, 47, 48–50] over the relative merits of different heterogeneity estimators.

In our simulations, levels of heterogeneity and effect size as well as $N_i$ and $k$ were based on empirical observations in psychology, which is a strength of our approach. We did not consider some heterogeneity estimators that previously showed promise e.g., the two-step PM estimator [51], and we did not systematically manipulate $N_i$, which is a factor of interest in itself [6, 37, 52]. These are limitations of our approach. However, the six factors manipulated here in conjunction with the five estimators we considered, already posed considerable challenges, both in terms of the simulations' run time as well as the ensuing analyses, and an even more complex simulation design would not have been feasible. Although our modelling of $p$-hacking was based on empirical observations [23], their implementation cannot avoid arbitrary choices. (For example, in our simulation only the result with the lowest $p$-value was submitted for publication. Other choices–e.g., the analysis that is based on the largest number of participants whilst obtaining $p < .05$ or on all analyses that obtain $p < .05$–would have been perfectly plausible.) Future studies will need to show how well our conclusions hold under modified assumptions. In this context it is encouraging that our simulation replicated key previous findings [28] even though our implementation of bias differed considerably from theirs. Finally, our simulations are restricted to continuous outcome measures, and it remains unclear if similar results are valid for binary outcome measures.

Meta-analyses in psychology often find large heterogeneity [1, 2]. More importantly, its causes typically remain unclear and this combination reflects poorly on the scientific understanding of the subject matter [15]. Our simulation results show that high observed heterogeneity cannot be conveniently dismissed as resulting from bias. For example, based on the DL estimator, average $T$ was found to be 0.33 in a large sample of meta-analyses in psychology [15]; our simulation found that DL rarely produces $T_{bias}$ even as high as 0.1. This underscores that, across many domains in psychology, large unaccounted heterogeneity is a serious issue that deserves more attention.

## Conclusion

For meta-analyses on continuous outcome measures we demonstrated here that the performance of heterogeneity estimators can differ considerably when effect sizes in the primary studies are distorted by publication bias and $p$-hacking, which is to be expected in many research domains [53–55]. Under various levels of distortions in the effect sizes of primary studies, heterogeneity estimates based on DL fared well in terms of bias and RMSE. However, as our own and previous work shows, REML outperforms DL in an unbiased research environment [6, 32]. Given that REML estimated heterogeneity almost as well as DL in a biased world (especially in simulation conditions that appear particularly plausible and/or important for actual research), REML remains in our view an excellent choice under the conditions simulated here, which should be broadly representative for meta-analyses of continuous outcomes in psychology.

For these conditions, our simulations suggest that the detrimental effects of biases in sets of primary studies are much larger for estimates of effect size than for estimates of heterogeneity. Therefore, our work underscores that the prevention and, in the case of past studies, detection and correction of biases in sets of primary studies is a pressing issue [10, 12, 22, 47–50, 56, 57].

## Supporting information

**S1 Fig. Mean bias in estimates of the true average effect size ($d_{\text{bias}}$) in the absence of publication bias and $p$-hacking for five heterogeneity estimators as a function of true average effect size ($\theta$), true heterogeneity ($\tau$), and number of studies per meta-analysis ($k$).** (TIF)

**S2 Fig. Coverage of 95% CIs around $d$ in the absence of publication bias and $p$-hacking for five heterogeneity estimators as a function of true heterogeneity ($\tau$) and number of studies per meta-analysis ($k$).** (TIF)

**S3 Fig. Standard deviation for heterogeneity estimates under constant simulation conditions in the absence of publication bias and $p$-hacking for five heterogeneity estimators as a function of true heterogeneity ($\tau$) and number of studies per meta-analysis ($k$).** (TIF)

**S4 Fig. Coverage of 95% CIs around $T$ in the absence of publication bias and $p$-hacking for the DL estimator as a function of true average effect size ($\theta$), true heterogeneity ($\tau$), and number of studies per meta-analysis ($k$).** Virtually identical results for other estimators not shown. (TIF)

**S5 Fig. Illustration of the strongest 3-way interaction on $T_{\text{bias}}$ (see Table 2).** (TIF)

**S6 Fig. Absence of interaction between effects of $p$-hacking and strength of publication bias on $T_{\text{bias}}$.** (TIF)

**S7 Fig. $P$-hacking and strength of publication bias differ in their interaction with the true average effect size ($\theta$) on $T_{\text{bias}}$.** (TIF)

**S8 Fig. Under 1-tailed publication bias (shown here), underestimation of heterogeneity is more prevalent than overestimation.** (TIF)

**S9 Fig. Illustration of the strongest 2-way interaction on $T_{\text{rmse}}$ (see Table 3).** (TIF)

**S10 Fig. Overestimation of effect size increases as heterogeneity increases.** (TIF)

## Acknowledgments

We would like to thank Thomas Pollet for helpful comments on an earlier draft.

## Author Contributions

**Conceptualization:** Johannes Hönekopp, Audrey Helen Linden.

**Data curation:** Audrey Helen Linden.

**Formal analysis:** Johannes Hönekopp.

**Software:** Johannes Hönekopp, Audrey Helen Linden.

**Validation:** Johannes Hönekopp, Audrey Helen Linden.

**Writing – original draft:** Johannes Hönekopp.

**Writing – review & editing:** Audrey Helen Linden.

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
