## [Decision Letter · Decision Letter 0]

24 Sep 2021

PONE-D-21-24961Heterogeneity estimates in a biased worldPLOS ONE

Dear Dr. Hönekopp,

Thank you for submitting your manuscript to PLOS ONE. After careful consideration, we feel that it has merit but does not fully meet PLOS ONE’s publication criteria as it currently stands. Therefore, we invite you to submit a revised version of the manuscript that addresses the points raised during the review process.

We look forward to receiving your revised manuscript.

Kind regards,

Tim Mathes

Academic Editor

PLOS ONE

Journal Requirements:

2. We note that you have referenced (ie. Bewick et al. [5]) which has currently not yet been accepted for publication. Please remove this from your References and amend this to state in the body of your manuscript: (ie “Bewick et al. [Unpublished]”) as detailed online in our guide for authors

Reviewers' comments:

Reviewer's Responses to Questions

**Comments to the Author**

1. Is the manuscript technically sound, and do the data support the conclusions?

Reviewer #1: Yes

Reviewer #2: Yes

Reviewer #3: Yes

2. Has the statistical analysis been performed appropriately and rigorously? 

Reviewer #1: Yes

Reviewer #2: Yes

Reviewer #3: Yes

3. Have the authors made all data underlying the findings in their manuscript fully available?

Reviewer #1: Yes

Reviewer #2: Yes

Reviewer #3: Yes

4. Is the manuscript presented in an intelligible fashion and written in standard English?

Reviewer #1: Yes

Reviewer #2: Yes

Reviewer #3: Yes

5. Review Comments to the Author

Reviewer #1: Dear Johannes and Audrey,

First of all, I’d like to complement you with your nice and relevant study. It was a joy to see that you’ve attempted to answer many remaining and highly relevant questions about heterogeneity, qrps and publication bias.

I will first provide you with my major comments. Minor comments and typos that caught my eye will be discussed later.

Kind regards,

Hilde Augusteijn

Major issues:

Method:

Page 9/page 24: It is unclear to me whether sample sizes were sampled from the total distribution of sample sizes from the 150 meta-analyses, or whether sample sizes were sampled per meta-analyses (e.g. randomly select 1 meta-analyses and generate data for all Ns in that meta-analyses). I suspect you did the former. I wonder however, how representative these sample sizes are within one meta-analysis. That is, in reality very small and very large studies may not be included in a meta-analyses together, since they do not investigate the same topics, or in ways that are methodically very different. Combining these small and large sample sizes might have a large impact on your heterogeneity estimates, as the variation of sample sizes matters for heterogeneity estimates (See Augusteijn et al, with a 1:1 ratio and 1:10 ratio). Please discuss the impact of your sample size choices in the discussion section.

Page 11: 1- and 2-tailed publication bias: Your interpretation of 1-tailed publication bias is different from how it is commonly interpreted in selection models. Your model of bias still has three different parameters: negative significant results: probability of publication is 0, non-significant results: probability depends on publication bias, positive significant results: probability of publication is 1. Often, non-significant and negative significant results are both considered to be effected by publication bias. Your choice will certainly have an impact on your results, especially when the true effect size is 0. Please change your publication bias model for 1-tailed bias, or provide a discussion of possible impact in your discussion, preferably with at least some additional simulations as a sensitivity analysis.

Page 11: Level of publication bias. This is my most important point of critique. I believe that your levels of publication bias a too limited. There are sufficient indications that publication bias in psychology might be higher than 80%. For example over 90% of studies report support for their focal hypothesis in the study by Fanelli (2010). Furthermore, the effects of publication bias on heterogeneity estimates are non-linear and heterogeneity estimates are impacted most drastically when publication bias is 100% (biggest underestimation), or close to 100 % (large overestimations when true effect is small). Please also include higher levels of publication bias. E.g. 90% or 95%, and 100%. Even though 100% bias might (luckily) not be realistic, neither is 0% bias. Knowing how the different estimators behave in this scenario is still highly relevant.

Minor issues:

Page 5: Not all QRPs are related to running multiple analyses and reporting only the smallest p-value. For example HARKing, rounding off p-values, or fraud. Furthermore, why did you choose these four? And do you expect they have exactly the same effect on the meta-analytical results, or not? What do we already know from previous studies on QRPs on for example effect size estimates?

Page 6: Do we know how often 2-tailed publication bias is plausible, compared to one tailed bias? Is there some data from empirical studies?

Page 9, line 189: please provide a reference for the claim that meta-analyses on continues outcomes are frequent.

Page 9, line 201: Is this median N of 100 the total N or per group?

Page 12, line 271: An I2 value of 6.6% when true tau=0, in conditions without qrps or bias. This deviates much more from 0 than I would expect?

Page 13, start of results: Please provide the reader with a sense of what the meta-analytical datasets looked like in the end: what was the effect of all qrps (splitting datasets, adding participants), on the actual sample sizes of primary studies? Is this still close to Ni=100?

Typos:

Page 5, line 111: QRPs instead of QRPS

Page 10, line 220: sections are not labeled as 2.2.

Page 13, line 288: URL to osf page no longer works. Please update the URL.

Page 15, line 347: ‘low k low’.

Page 24, line 500: ‘were’ instead of ‘where’.

Reviewer #2: “Heterogeneity estimates in a biased world” is a Monte Carlo study of the effects of publication bias (PB) and QRP (questionable research practices). It seems to be rigorously conducted, and its simulations are based on realistic research conditions as seen in psychology. Its major findings is: “Our results showed that biases in primary studies caused much greater problems for the estimation of effect size than for the estimation of heterogeneity.” This is an important lesson that meta-analysis community needs to hear. I suspect that this was already widely known, but I believe that this is the first paper that demonstrates this is a clear, rigorous and replicable way.

I wish to congratulate the authors for the way they conduct their Monte Carlo simulations. The design of the simulations can make an enormous difference to their results. Unless, the important research parameters (sample sizes, the amount of heterogeneity, the degree of publication selection, etc.) accurately reflects what is seen in the actual relevant research literature, the findings will be largely irrelevant. However, the authors based their simulations on what they found in what seems to be a fairly representative sample of 150 meta-analyses in psychology. I recommend that PLOS published this paper with a few revisions.

Suggestions for revisions.

1. Emphasize main findings: Please emphasize and expand the main finding that it is PB and QRP that causes random effect (RE) to be so very bias, and that this bias is very large under the typical conditions that the authors simulate. This substantial bias has also been confirmed in a systematic review of large pre-registered multi-lab replications (Kvarven et al., 2020), and it is so large that RE is entirely unreliable if applied to psychology naively without many qualifications and auxiliary statistical checks. These biases are also of a notable scientific size. The authors need to state a bit more strongly how these different methods of estimating tau (the heterogeneity SD) are largely irrelevant, especially relative to the size and consequences of RE’s bias. These consequences need to be explicitly stated and emphasized.

2. Biased studies: The way the authors characterize PB and QRP is rather misleading and may give the broader audience the wrong impression about the nature and extent of the problems involved. Classical PB is itself often interpreted as merely omitting some studies that are not statistically significant (SS). While this is indeed one avenue for the bias that we often find in published research results, there are many others. Reporting bias is recognized as a different avenue by medical researchers, as QRP is recognized by psychologists. But all of these vectors of the biases are the result of some process of selection of the results to be SS. This selection can be undertaken by the researchers on their own for their own reasons, or in anticipation of what reviewers and editors might demand. Or, this selection can be forced by the reviewers and editors. These details of selection process are largely irrelevant because they have the same outcome and can be simulated in the same way. Thus, a little more discussion of what this bias is and more references to the classical and better regraded methods to correct for these biases (collectively called PB here, for short) is needed. The authors repeatedly characterize this problem as “biases in primary studies.” It is not, or at least, this is not necessarily a bias in the primary studies. PB can be very serious, just as we see it in practice, if the individual primary results are not biased, but merely were selectively reported to be SS from entirely randomly produced distribution of estimates (with random QRPs, random outcome measures, random samples, etc). You might say that PB is an emergent property (selection for SS) of the entire research literature in a given area but is not associated with individually biased studies. Studies and researchers may also be biased, which will only amplify PB, but focusing on the unnecessary bias of individual studies can cause many to dismiss this severe problem. Many researchers do not believe that a notable portion of their colleagues are dishonest or deliberately distorting science. This is why, PB is so pernicious and easily dismissed. It can emerge from the system, as a whole, without individually researchers knowingly distorting science. Please do not characterize PB as ‘biased studies’ but rather as studies selected to be SS.

3. Type I errors: Please report the type I errors of all of these methods using the current simulation design. I suspect that the authors will find that RE has very high rates of type I errors for all of these methods, at least as long as there are more than a few estimates. If so, this will confirm the systematic review of large pre-registered multi-lab replications (Kvarven et al., 2020). Rates of false positives are very important as an indicator of scientific credibility. I suspect that RE (regardless of the method use to estimate tau) has such high rates of false positives, using the authors’ current simulation design, to disqualify RE from any serious scientific use. In any case, type I errors are important to show and to discuss. Not reporting type I errors could be considered to be a type of a selection bias in the way these simulation results are displayed and published. Methods PB, if you will.

4. Alternatives to random effects: This entire study assumes that RE is the only adequate method to conduct basic meta-analysis in psychology and that this issue then comes down to the best way to calculate RE. This is not the case, and worse, the authors show that all the ways to calculate RE produce notably large bias (greatly exaggerating the size of the effect under examination). I suspect that this simulation design will show that RE has high rates of false positives. It has long been known that RE has unacceptable biases and that these biases are easily reduced (Henmi and Copas, 2010; Stanley and Doucouliagos, 2015). Henmi and Copas (2010) showed that FE (fixed effect) will notably reduces PB and that the RE’s estimate of tau can accommodate the heterogeneity that FE ignores. However, Henmi and Copas (2010) uses the DL estimate of tau in their calculation of the CI. So, the estimate of tau might still be important in their approach. Henmi and others (2021) has recently generalized this method and show how it can work for very small meat-analyses. Alternatively, an entirely different approach, the unrestricted weighted least squares (UWLS), uses the bias reduction of FE but automatically accommodates heterogeneity using the mathematical invariance of WLS’s variance-covariance matrix to any multiplicative constant. UWLS accommodates heterogeneity without referring to or using RE or any of its estimates of tau (Stanley and Doucouliagos, 2015; 2017). That is, the central issue of this study of the effect of PB on estimates of tau could be entirely avoided and, at the same time, reduce the large biases reported in this paper. Simulations, like these, have shown that UWLS notably reduces RE’s bias with little if any compensating statistical loss (Stanley and Doucouliagos, 2015; 2017). These alternative methods to RE have been widely applied across the disciplines and used as a basis for a new statistical method to detect PB (Stanley et al., 2021). It would be nice if these other methods were simulated and reported using this same design. At a minimum discussed, they need to be discussed as viable alternative to this concern about how tau is calculated and as an alternative to RE’s large biases and high rates of false positives. The central scientific question, is how to reduce or eliminate bias and false positive meta-analyses because they are often the best scientific evidence we have.

References:

Henmi M, Copas JB. Confidence intervals for random effects meta-analysis and robustness to publication bias. Statistics in Medicine, 2010; 29:2969–2983.

Henmi M, Hattori S, Friede T. A confidence interval robust to publication bias for random-effects meta-analysis of few studies. Res Syn Meth. 2021;12:674–679. https://doi.org/10.1002/jrsm.1482

Stanley, T.D. and Doucouliagos, C. Neither fixed nor random: Weighted least squares meta-analysis,” Statistics in Medicine, 2015: 342116-27.

Stanley, T.D. and Doucouliagos, C. Neither fixed nor random: Weighted least squares meta-regression analysis. Res Synth Methods. 2017;8:19-42.

Stanley TD, Doucouliagos H, Ioannidis JPA, Carter EC. Detecting publication selection bias through excess statistical significance. Research Synthesis Methods. 2021; 1-20. https://doi.org/10.1002/jrsm.1512

Reviewer #3: Review comments to the author can be found in the attached .docx document. They are organised in the sequence of the paper and include some general points and more specific questions to be responded to.

6. PLOS authors have the option to publish the peer review history of their article (what does this mean?). If published, this will include your full peer review and any attached files.

Reviewer #1: **Yes: **Hilde Augusteijn

Reviewer #2: No

Reviewer #3: No

---

## [Author Response · Author response to Decision Letter 0]

1 Nov 2021

Authors: Thanks for your thorough reading of our paper. We appreciate your detailed feedback and found your comments helpful.

I enjoyed reading this simulation study examining the effect of bias on heterogeneity estimates in meta-analysis. I found the methodology and inferences drawn from the results to be broadly sound. I have some minor suggestions which I have documented below.

It is very pleasant to see implementation of open science requirements. Thank-you for making all your code easily available. However I was unable to run the R files to replicate due to the absence of the excel file “obsN.xlsx”

Authors: Sorry about that. The file is now uploaded.

Line 59 “is an example for heterogeneity.” - possibly is an example of heterogeneity?

Authors: Corrected.

Line 95 “Bias in primary studies” Perhaps this is an accepted term in the psychology literature, but I would argue that publication bias arises from aggregation of primary studies but does not imply bias in any individual study result. That would make calling it a bias in a primary study potentially confusing or misleading. This term recurs throughout the paper and I wonder if some alternative terminology would improve the clarity of the paper. Perhaps more clearly you might call them sources of bias in meta-analysis.

Authors: Our use of the expression QRPs in our original submission caused some confusion. This section covers the two motivationally related but conceptually distinct processes of (i) publication bias and (ii) flexibility in data collection and analysis (now “p-hacking”, previously QRPs). P-hacking might cause systematic bias in individual studies. Motivated by your comment, we changed the heading to “Bias in published primary studies”.

Line 97 I completely agree with your point here that both publication bias and QRPs are sources of bias but this binary categorisation is non-standard in my experience and I think potentially confusing. Questionable research practices is a deliberately broad term often used as a catch all for research misconduct short of outright fraud/falsification of data. In my experience it normally isn’t associated with a particular mechanism or pattern of bias as you write about multiple analyses (though I would be happy to be corrected with an appropriate reference). In addition publication bias can conceivably be caused by questionable research practices (e.g. not seeking publication for null results). I think it’s fine to keep using the term throughout the paper but perhaps if you altered the section introducing these biases to reduce the emphasis on these as the only two sources of bias, the complete separation of the two, and clarify that the QRPs you want to focus on are only a subset of a larger group of practices.

Authors: Our use of the expression QRPs was unfortunate, and we replaced it with “p-hacking” throughout, which should clarify the matter.

Line 113 This reference shows that the listed QRPs are commonly self reported, however it would be great to know if there was any empirical evidence for their effects introducing meta-analysis results if you are aware of any. For an example of this from the medical field see:

Savović J, Jones HE, Altman DG, Harris RJ, Jüni P, Pildal J, et al. Influence of Reported Study Design Characteristics on Intervention Effect Estimates From Randomized, Controlled Trials. Annals of Internal Medicine. 2012 Sep 18;157(6):429.

Authors: The frequently self-reported p-hacking strategies we focused on are not easily identifiable from study descriptions. We are not aware of any studies in the spirit of Savović et al. that empirically investigate the bias caused by our (or similar) p-hacking strategies. 

Line 189 Consider citing metafor

Authors: Done

Line 208 “Imagine that a given level of publication bias and QRPs led to a bias of 0.1 in the overall effect size estimate 𝜃̂ and a bias of 0.1 in the heterogeneity estimate 𝜏̂. In this case it would be sensible to conclude that effect size estimates and heterogeneity estimates were affected to the same extent.” I think this needs either further justification, or more ambiguity. Whilst it might be narrowly true that they have the same numerical level of bias, what the implications of this are is far from clear. Bias in estimate of effect are likely to affect heterogeneity, and the estimate of effect tends to be the focus of systematic review, so will typically have a larger effect on interpretation though of course context is important.

Authors: We now qualify our statement, and the sentence ends with “(although the same degree of bias might be seen as more consequential for effect size estimates than for heterogeneity estimates).”, see Line 209.

Line 229 “Optional outlier removal: Researchers run separate analyses on all data, and on data with 

outliers (unsigned z ≥ 2) removed.” This doesn’t seem a likely mechanism – surely only outliers which push away from significance would be excluded (though I appreciate this is different assuming a 2 tail publication bias).

Authors: Depending on the simulation parameters, simulated studies are analysed with and without outliers. Reporting and publication are made contingent on the resulting p-values. Thus, outliers that lower the p-value will be (perhaps implausibly) removed for the analysis without outliers; however, their removal leads to a larger p-value, which is why this result will not be reported and therefore will not enter the meta-analysis/the simulation results. 

Line 235 or 252 “Under 1-tailed publication bias, results that went against the expected

direction were never published” This choice makes the 1 tailed bias much more aggressive than the 2 tailed, and perhaps a bit unrealistic in the modern publishing environment? There is good evidence for publication bias on p-value, but I am aware of less evidence for publication bias based on direction of effect (though obviously they are related). It would be useful to see more justification for the formulations of publication bias chosen here, particularly any empirical evidence for the levels chosen.

Authors: We modified our implementation of 1-tailed publication bias (PB) in light of your comments. We re-ran all simulations with 1-tailed PB. Under 1-tailed PB, statistically significant results (2-tailed testing) in the expected direction were always published; all other results were censored to a degree that was defined by the strength of PB. This is in line with Augusteijn et al., 2019. If p-hacking required selection between multiple analyses, this was contingent on a modified p-value, which equaled p for results in the expected direction. For results in the opposite direction, the modified p-value was computed as 1 + (1-p). Obviously, being >1 the modified p-value cannot be interpreted as a probability, but it appropriately penalizes results in the wrong direction with, ceteris paribus, stronger effects carrying greater penalties. (See Lines 254-261). All analyses and figures in this revision are based on this new version of 1-tailed PB. Note that results and conclusions did not change in substantive ways.

Line 269 “ we computed mean I 2 levels across” I would be more interested in median I2 values if these are easily computable from your results

Authors: We added the medians in brackets, see Line 278.

Line 279 “Following (10), 1,000 meta-analyses were run for each” I appreciate the computational considerations here, but did you re-run an evaluation of the monte-carlo error for this simulation, or did you use 1,000 repetitions because that was sufficient in the previous paper? It is possible to estimate the Monte Carlo Error without running a much larger simulation as per:

Koehler E, Brown E, Haneuse SJ-PA. On the Assessment of Monte Carlo Error in Simulation-Based Statistical Analyses. Am Stat. 2009 May 1;63(2):155–62.

Authors: For each cell in the design, the simulation computed and recorded the standard deviation across the 1,000 heterogeneity estimates. This allowed us to estimate the Monte Carlo Error (MCE) for each cell. Mean MCE was 0.0023 with a maximum of 0.0068, which strikes us as satisfactory (see Lines 291-295, where we also refer to the Koehler et al. reference). Unfortunately, the standard deviation for effect size estimates was not recorded. Therefore, we could not estimate its MCE. 

In addition did you consider using parallel computing to improve the speed of computation? I haven’t been able to exactly replicate your analysis but I have recently used R packages such as foreach and doParallel to run simulations on multiple cores without much programming difficulty to substantially improve running times.

Authors: Thanks for the tip. We will look into this in future simulations.

Line 293 “ level of effect size proved of little consequence” It might be worth clarifying somewhere that this is only true for the mean difference, for other measures (such as odds ratio) it can make a difference.

Authors: We address this among the limitations of our study (Line 560-562). 

Line 336 In S4 fig is the coverage for all estimates of heterogeneity combined or for a specific estimate (e.g. DL)

Authors: Somewhat counterintuitively, metafor confidence intervals for heterogeneity are independent of the heterogeneity estimator. Therefore, CIs are the same for all five heterogeneity estimators. 

For all figures please consider using a scalable format (e.g. pdf/svg/eps) rather than png so that there is no loss of resolution when zooming in to differentiate between lines. In addition in some figures (e.g. figure 3 tau 0.44 segment) it is very difficult to establish which lines overlap. Perhaps you could consider colour, or alternative types of lines?

Authors: You will find that all figures are much bigger and clearer now, although we saved them in the TIFF format as recommended by the journal.

I would like to see a little more justification of the reliance on the ANOVA here and in other places in the paper. It seems appropriate to infer that stability in estimates in the presence of bias is desirable, but absolute levels of bias and RMSE may be more valuable in certain situations.

Authors: We agree that a sizable or even large level of bias or RMSE is relevant even if it remains constant across conditions. Our figures address these levels of bias/RMSE, and our (necessarily somewhat arbitrary) verbal labels (Line 215) should also help to focus on this issue. 

However, it is also important to consider how bias/RMSE varies as a function of the factors manipulated in the simulation. Given the complexity of our design, ANOVA is just a convenient tool to draw attention to powerful factors and interactions. Similarly, ANOVA results demonstrated that higher-order interactions (i.e., more than 2-way) were typically of little importance, which protects authors and readers against being sidetracked. 

Line 374 Table 2. Does “M” indicate the mean? Please clarify this

Authors: It does. Now clarified in the Table’s note, Line 388.

Figure 4: Effect size, type of PB, and QRP environment are all drawn as categorical variables (though effect size could be redrawn on a continuous scale) and so it may not be appropriate to draw lines between the point values.

Authors: Your comment is, of course, correct. However, in psychology lines are used even for categorical independent variables to facilitate the perception of interaction effects. Here, we keep with this tradition. 

I think this section (and others) would also benefit from a deeper investigation of the difference in patterns between PB and QRPs. Implementing QRPs is a novelty of the paper, and I think a more in depth comparison of the effects of PB and QRPs would be interesting.

Authors: We now provide more detail (Lines 430-439).

Line 383 “Overall, T bias was driven upwards ...” This is true and the trends are obvious from the figure, but since bias is optimal close to zero it might be worth rewording this to make it clear when bias becomes worse (i.e. further from zero) rather than simply higher – which could be better if the starting point was negative bias.

Authors: The remainder of the paragraph describes the resulting biases in greater detail (Lines 396-404).

Line 386 “This meant that the highest levels of T bias ...” Highest on the absolute scale, but I believe highest is technically incorrect here. This is related to the point above.

Authors: We rephrased to “the largest positive levels of Tbias” (Line 398).

Line 397 “Regarding 2-way interactions, effect size ⨯ type of publication bias ...” You could consider mentioning potential mechanisms of why bias is high where they are obvious to you.

Authors: For Fig 5, we added the following explanation (Line 414): “This arises because only under 2-tailed publication bias do p-hacking and publication bias have the potential to push published effect sizes either above or below zero, thus maximising their variance.” For Fig 6, we added (Line 420), “This reflects that strong publication bias maximises the variance in published effect sizes at θ = 0 because (exaggerated) published effect sizes are equally likely to be above or below zero.”

Line 416 “is less prevalent than underestimation” I’m not sure you can comment on prevalence since you are not doing empirical work – it only occurs more commonly in these simulation conditions. This is also another situation where perhaps some mechanistic explanation might orientate the reader.

Authors: We rephrased to, “overestimation of heterogeneity occurs under fewer simulation conditions is less prevalent than underestimation” (Line 442).

Line 450 This section feels a bit minimalist. Effect size is often the main focus of a paper, and is given relatively little attention here. One thing to consider is giving an example of the magnitude of bias in the text when comparing publication bias and QRPs. I think this is worth noting since 

Authors: We now provide some perspective on the magnitude of bias in effect size estimates, “The combination of [p-hacking and publication bias] could induce large bias. E.g., for a true effect size of zero, θ might be estimated to be over 0.3, a substantial effect” (Line 487). Part of your comment got lost. We hope this addresses the issue.

Line 485 “and also those with θ=0 (because such effects tend to be of little interest to researchers” I’m not sure this is true, especially since researchers don’t know the true population effect in advance.

Authors: We now express this point more carefully, see Lines 507-513.

Line 492 “From this perspective, publication bias and QRPs cause much more problems for the estimation of effect size than for the estimation of heterogeneity” It might be worth exploring graphically representing the distribution of bias for t and d (e.g. histogram/boxplot/violin plot) rather than giving proportions above arbitrary cut-offs.

Authors: We ditched the proportions and now provide boxplots instead (Fig9, Line 506).

Line 522 “we did not consider some heterogeneity estimators that previously showed promise” It would be helpful to give an example here

Authors: We added a reference (Line 548).

---

## [Editor Report · Decision Letter 1]

11 Nov 2021

PONE-D-21-24961R1Heterogeneity estimates in a biased worldPLOS ONE

Dear Dr. Hönekopp,

Thank you for submitting your manuscript to PLOS ONE. After careful consideration, we feel that it has merit but does not fully meet PLOS ONE’s publication criteria as it currently stands. Therefore, we invite you to submit a revised version of the manuscript that addresses the points raised during the review process. The comments of reviewer 1 and 2 have not been addressed. Please address all comments that were raised by the reviewers.

We look forward to receiving your revised manuscript.

Kind regards,

Tim Mathes

Academic Editor

PLOS ONE
---

## [Author Response · Author response to Decision Letter 1]

16 Dec 2021

Reviewer #3

Our response: Thanks for your thorough reading of our paper. We appreciate your detailed feedback and found your comments helpful. Please note that all line numbers in our reply refer to the clean copy, i.e., the copy without tracked changes.

I enjoyed reading this simulation study examining the effect of bias on heterogeneity estimates in meta-analysis. I found the methodology and inferences drawn from the results to be broadly sound. I have some minor suggestions which I have documented below.

It is very pleasant to see implementation of open science requirements. Thank-you for making all your code easily available. However I was unable to run the R files to replicate due to the absence of the excel file “obsN.xlsx”

Our response: Sorry about that. The file is now uploaded.

Line 59 “is an example for heterogeneity.” - possibly is an example of heterogeneity?

Our response: Corrected.

Line 95 “Bias in primary studies” Perhaps this is an accepted term in the psychology literature, but I would argue that publication bias arises from aggregation of primary studies but does not imply bias in any individual study result. That would make calling it a bias in a primary study potentially confusing or misleading. This term recurs throughout the paper and I wonder if some alternative terminology would improve the clarity of the paper. Perhaps more clearly you might call them sources of bias in meta-analysis.

Our response: Wherever appropriate, we changed “biased primary studies” or similar to “biased sets of primary studies” or similar.

Line 97 I completely agree with your point here that both publication bias and QRPs are sources of bias but this binary categorisation is non-standard in my experience and I think potentially confusing. Questionable research practices is a deliberately broad term often used as a catch all for research misconduct short of outright fraud/falsification of data. In my experience it normally isn’t associated with a particular mechanism or pattern of bias as you write about multiple analyses (though I would be happy to be corrected with an appropriate reference). In addition publication bias can conceivably be caused by questionable research practices (e.g. not seeking publication for null results). I think it’s fine to keep using the term throughout the paper but perhaps if you altered the section introducing these biases to reduce the emphasis on these as the only two sources of bias, the complete separation of the two, and clarify that the QRPs you want to focus on are only a subset of a larger group of practices.

Our response: Our use of the expression QRPs was unfortunate, and we replaced it with “p-hacking” throughout, which should clarify the matter.

Line 113 This reference shows that the listed QRPs are commonly self reported, however it would be great to know if there was any empirical evidence for their effects introducing meta-analysis results if you are aware of any. For an example of this from the medical field see:

Savović J, Jones HE, Altman DG, Harris RJ, Jüni P, Pildal J, et al. Influence of Reported Study Design Characteristics on Intervention Effect Estimates From Randomized, Controlled Trials. Annals of Internal Medicine. 2012 Sep 18;157(6):429.

Our response: The frequently self-reported p-hacking strategies we focused on are not easily identifiable from study descriptions. We are not aware of any studies in the spirit of Savović et al. that empirically investigate the bias caused by our (or similar) p-hacking strategies. 

Line 189 Consider citing metafor

Our response: Done

Line 208 “Imagine that a given level of publication bias and QRPs led to a bias of 0.1 in the overall effect size estimate 𝜃̂ and a bias of 0.1 in the heterogeneity estimate 𝜏̂. In this case it would be sensible to conclude that effect size estimates and heterogeneity estimates were affected to the same extent.” I think this needs either further justification, or more ambiguity. Whilst it might be narrowly true that they have the same numerical level of bias, what the implications of this are is far from clear. Bias in estimate of effect are likely to affect heterogeneity, and the estimate of effect tends to be the focus of systematic review, so will typically have a larger effect on interpretation though of course context is important.

Our response: We now qualify our statement, and the sentence ends with “(although the same degree of bias might be seen as more consequential for effect size estimates than for heterogeneity estimates).”, see Line 217.

Line 229 “Optional outlier removal: Researchers run separate analyses on all data, and on data with 

outliers (unsigned z ≥ 2) removed.” This doesn’t seem a likely mechanism – surely only outliers which push away from significance would be excluded (though I appreciate this is different assuming a 2 tail publication bias).

Our response: Depending on the simulation parameters, simulated studies are analysed with and without outliers. Reporting and publication are made contingent on the resulting p-values. Thus, outliers that lower the p-value will be (perhaps implausibly) removed for the analysis without outliers; however, their removal leads to a larger p-value, which is why this result will not be reported and therefore will not enter the meta-analysis/the simulation results. 

Line 235 or 252 “Under 1-tailed publication bias, results that went against the expected

direction were never published” This choice makes the 1 tailed bias much more aggressive than the 2 tailed, and perhaps a bit unrealistic in the modern publishing environment? There is good evidence for publication bias on p-value, but I am aware of less evidence for publication bias based on direction of effect (though obviously they are related). It would be useful to see more justification for the formulations of publication bias chosen here, particularly any empirical evidence for the levels chosen.

Our response: We modified our implementation of 1-tailed publication bias (PB) in light of your comments. We re-ran all simulations with 1-tailed PB. Under 1-tailed PB, statistically significant results (2-tailed testing) in the expected direction were always published; all other results were censored to a degree that was defined by the strength of PB. This is in line with Augusteijn et al., 2019. If p-hacking required selection between multiple analyses, this was contingent on a modified p-value, which equaled p for results in the expected direction. For results in the opposite direction, the modified p-value was computed as 1 + (1-p). Obviously, being >1 the modified p-value cannot be interpreted as a probability, but it appropriately penalizes results in the wrong direction with, ceteris paribus, stronger effects carrying greater penalties. (See Lines 260-268). All analyses and figures in this revision are based on this new version of 1-tailed PB. Note that results and conclusions did not change in substantive ways.

Line 269 “ we computed mean I 2 levels across” I would be more interested in median I2 values if these are easily computable from your results

Our response: We added the medians in brackets, see Line 285.

Line 279 “Following (10), 1,000 meta-analyses were run for each” I appreciate the computational considerations here, but did you re-run an evaluation of the monte-carlo error for this simulation, or did you use 1,000 repetitions because that was sufficient in the previous paper? It is possible to estimate the Monte Carlo Error without running a much larger simulation as per:

Koehler E, Brown E, Haneuse SJ-PA. On the Assessment of Monte Carlo Error in Simulation-Based Statistical Analyses. Am Stat. 2009 May 1;63(2):155–62.

Our response: For each cell in the design, the simulation computed and recorded the standard deviation across the 1,000 heterogeneity estimates. This allowed us to estimate the Monte Carlo Error (MCE) for each cell. Mean MCE was 0.0023 with a maximum of 0.0068, which strikes us as satisfactory (see Lines 297-301, where we also refer to the Koehler et al. reference). Unfortunately, the standard deviation for effect size estimates was not recorded. Therefore, we could not estimate its MCE. 

In addition did you consider using parallel computing to improve the speed of computation? I haven’t been able to exactly replicate your analysis but I have recently used R packages such as foreach and doParallel to run simulations on multiple cores without much programming difficulty to substantially improve running times.

Our response: Thanks for the tip. We will look into this in future simulations.

Line 293 “ level of effect size proved of little consequence” It might be worth clarifying somewhere that this is only true for the mean difference, for other measures (such as odds ratio) it can make a difference.

Our response: We address this among the limitations of our study (Line 584-586). 

Line 336 In S4 fig is the coverage for all estimates of heterogeneity combined or for a specific estimate (e.g. DL)

Our response: Somewhat counterintuitively, metafor confidence intervals for heterogeneity are independent of the heterogeneity estimator. Therefore, CIs are the same for all five heterogeneity estimators. 

For all figures please consider using a scalable format (e.g. pdf/svg/eps) rather than png so that there is no loss of resolution when zooming in to differentiate between lines. In addition in some figures (e.g. figure 3 tau 0.44 segment) it is very difficult to establish which lines overlap. Perhaps you could consider colour, or alternative types of lines?

Our response: You will find that all figures are much bigger and clearer now, although we saved them in the TIFF format as recommended by the journal.

I would like to see a little more justification of the reliance on the ANOVA here and in other places in the paper. It seems appropriate to infer that stability in estimates in the presence of bias is desirable, but absolute levels of bias and RMSE may be more valuable in certain situations.

Our response: We agree that a sizable or even large level of bias or RMSE is relevant even if it remains constant across conditions. Our figures address these levels of bias/RMSE, and our (necessarily somewhat arbitrary) verbal labels (Line 222) should also help to focus on this issue. 

However, it is also important to consider how bias/RMSE varies as a function of the factors manipulated in the simulation. Given the complexity of our design, ANOVA is just a convenient tool to draw attention to powerful factors and interactions. Similarly, ANOVA results demonstrated that higher-order interactions (i.e., more than 2-way) were typically of little importance, which protects authors and readers against being sidetracked. 

Line 374 Table 2. Does “M” indicate the mean? Please clarify this

Our response: It does. Now clarified in the Table’s note, Line 397.

Figure 4: Effect size, type of PB, and QRP environment are all drawn as categorical variables (though effect size could be redrawn on a continuous scale) and so it may not be appropriate to draw lines between the point values.

Our response: Your comment is, of course, correct. However, in psychology lines are used even for categorical independent variables to facilitate the perception of interaction effects. Here, we keep with this tradition. 

I think this section (and others) would also benefit from a deeper investigation of the difference in patterns between PB and QRPs. Implementing QRPs is a novelty of the paper, and I think a more in depth comparison of the effects of PB and QRPs would be interesting.

Our response: We now provide more detail (Lines 440-449).

Line 383 “Overall, T bias was driven upwards ...” This is true and the trends are obvious from the figure, but since bias is optimal close to zero it might be worth rewording this to make it clear when bias becomes worse (i.e. further from zero) rather than simply higher – which could be better if the starting point was negative bias.

Our response: The remainder of the paragraph describes the resulting biases in greater detail (Lines 405-413).

Line 386 “This meant that the highest levels of T bias ...” Highest on the absolute scale, but I believe highest is technically incorrect here. This is related to the point above.

Our response: We rephrased to “the largest positive levels of Tbias” (Line 407).

Line 397 “Regarding 2-way interactions, effect size ⨯ type of publication bias ...” You could consider mentioning potential mechanisms of why bias is high where they are obvious to you.

Our response: For Fig 5, we added the following explanation (Line 423): “This arises because only under 2-tailed publication bias do p-hacking and publication bias have the potential to push published effect sizes either above or below zero, thus maximising their variance.” For Fig 6, we added (Line 429), “This reflects that strong publication bias maximises the variance in published effect sizes at θ = 0 because (exaggerated) published effect sizes are equally likely to be above or below zero.”

Line 416 “is less prevalent than underestimation” I’m not sure you can comment on prevalence since you are not doing empirical work – it only occurs more commonly in these simulation conditions. This is also another situation where perhaps some mechanistic explanation might orientate the reader.

Our response: We rephrased to, “overestimation of heterogeneity occurs under fewer simulation conditions is less prevalent than underestimation” (Line 452).

Line 450 This section feels a bit minimalist. Effect size is often the main focus of a paper, and is given relatively little attention here. One thing to consider is giving an example of the magnitude of bias in the text when comparing publication bias and QRPs. I think this is worth noting since 

Our response: We now provide some perspective on the magnitude of bias in effect size estimates, “The combination of [p-hacking and publication bias] could induce large bias. E.g., for a true effect size of zero, θ might be estimated to be over 0.3, a substantial effect” (Line 497). Part of your comment got lost. We hope this addresses the issue.

Line 485 “and also those with θ=0 (because such effects tend to be of little interest to researchers” I’m not sure this is true, especially since researchers don’t know the true population effect in advance.

Our response: We now express this point more carefully, see Lines 530-534.

Line 492 “From this perspective, publication bias and QRPs cause much more problems for the estimation of effect size than for the estimation of heterogeneity” It might be worth exploring graphically representing the distribution of bias for t and d (e.g. histogram/boxplot/violin plot) rather than giving proportions above arbitrary cut-offs.

Our response: We ditched the proportions and now provide boxplots instead (Fig10, Line 533).

Line 522 “we did not consider some heterogeneity estimators that previously showed promise” It would be helpful to give an example here

Our response: We added a reference (Line 572).

Reviewer #1: Dear Johannes and Audrey,

First of all, I’d like to complement you with your nice and relevant study. It was a joy to see that you’ve attempted to answer many remaining and highly relevant questions about heterogeneity, qrps and publication bias.

I will first provide you with my major comments. Minor comments and typos that caught my eye will be discussed later.

Kind regards,

Hilde Augusteijn

Our response: Dear Hilde, thanks for your comments and suggestions. We appreciate your thoughts. Sorry that we failed to notice them when we prepared our first revision. Please note that all line numbers in our reply refer to the clean copy, i.e., the copy without tracked changes.

Major issues:

Method:

Page 9/page 24: It is unclear to me whether sample sizes were sampled from the total distribution of sample sizes from the 150 meta-analyses, or whether sample sizes were sampled per meta-analyses (e.g. randomly select 1 meta-analyses and generate data for all Ns in that meta-analyses). I suspect you did the former. I wonder however, how representative these sample sizes are within one meta-analysis. That is, in reality very small and very large studies may not be included in a meta-analyses together, since they do not investigate the same topics, or in ways that are methodically very different. Combining these small and large sample sizes might have a large impact on your heterogeneity estimates, as the variation of sample sizes matters for heterogeneity estimates (See Augusteijn et al, with a 1:1 ratio and 1:10 ratio). Please discuss the impact of your sample size choices in the discussion section.

Our response: To clarify the matter, we rephrased as follows: “We aggregated observed sample sizes from a representative set of 150 psychological meta-analyses (15) into a single distribution. Sample sizes Ni for simulated studies were randomly sampled from this distribution and equally split between groups 1 and 2.” (LL198). We addressed your concern about unrealistic combinations of very small and very large studies in the same simulated meta-analyses as follows: “If average sample size differed considerably across the 150 meta-analyses in our set, our approach might result in unrealistic combinations of very large and very small samples in simulated meta-analyses, which in turn might distort our results (Augusteijn, van Aert, & van Assen, 2019). However, an ANOVA (bias corrected accelerated bootstrap with 1,000 samples) revealed little variation of average sample size across these 150 meta-analyses (ηp2 = 0.020, F(149, 7077) = 0.97, p = .595).” See Lines 198-206. 

Page 11: 1- and 2-tailed publication bias: Your interpretation of 1-tailed publication bias is different from how it is commonly interpreted in selection models. Your model of bias still has three different parameters: negative significant results: probability of publication is 0, non-significant results: probability depends on publication bias, positive significant results: probability of publication is 1. Often, non-significant and negative significant results are both considered to be effected by publication bias. Your choice will certainly have an impact on your results, especially when the true effect size is 0. Please change your publication bias model for 1-tailed bias, or provide a discussion of possible impact in your discussion, preferably with at least some additional simulations as a sensitivity analysis.

Our response: We modified our implementation of 1-tailed publication bias (PB) in light of reviewers’ comments. We re-ran all simulations with 1-tailed PB. Under 1-tailed PB, statistically significant results (2-tailed testing) in the expected direction were always published; all other results were censored to a degree that was defined by the strength of PB. This is in line with Augusteijn et al., 2019. If p-hacking required selection between multiple analyses, this was contingent on a modified p-value, which equaled p for results in the expected direction. For results in the opposite direction, the modified p-value was computed as 1 + (1-p). Obviously, being >1 the modified p-value cannot be interpreted as a probability, but it appropriately penalizes results in the wrong direction with, ceteris paribus, stronger effects carrying greater penalties. (See Lines 261-268). All analyses and figures in this revision are based on this new version of 1-tailed PB. Note that results and conclusions did not change in substantive ways.

Page 11: Level of publication bias. This is my most important point of critique. I believe that your levels of publication bias a too limited. There are sufficient indications that publication bias in psychology might be higher than 80%. For example over 90% of studies report support for their focal hypothesis in the study by Fanelli (2010). Furthermore, the effects of publication bias on heterogeneity estimates are non-linear and heterogeneity estimates are impacted most drastically when publication bias is 100% (biggest underestimation), or close to 100 % (large overestimations when true effect is small). Please also include higher levels of publication bias. E.g. 90% or 95%, and 100%. Even though 100% bias might (luckily) not be realistic, neither is 0% bias. Knowing how the different estimators behave in this scenario is still highly relevant.

Our response: You raise an interesting point. We are less pessimistic about the prevalence of publication bias (PB) than you. The reason is that PB should predominantly affect studies’ focal hypothesis. Naturally, meta-analyses (MAs) also include results that were not the focal hypothesis of the paper in which they were published. These “non-headline” results should be less affected by PB, if at all. Two empirical observations support our viewpoint. 1) Frequently, a substantial proportion of primary effects summarised in a MA are not statistically significant as MAs' forest plots reveal. We are not aware of a systematic investigation of this issue but point to two arbitrary examples, Macnamara, Hambrick, and Oswald (2014, see Fig. 2) and (Sisk, Burgoyne, Sun, Butler, & Macnamara, 2018, see Fig. 2). 2) Levine, Asada, and Carpenter (2009) looked at the correlation between effect size and sample size across 51 meta-analyses. They found a much weaker correlation (mean r = -.16) than Kühberger, Fritz, and Scherndl (2014) who investigated the same relationship in findings that constituted the focal hypothesis of the respective papers and found rS = -.45.

Naturally, choices for other PB levels than ours would be perfectly defensible, and the study of additional PB levels would add to our simulation. However, we believe that the levels we chose are illuminating and sensible. This includes 0% PB, which we would expect in pre-registered trials and perhaps for some questions that are based on data irrelevant to papers’ focal hypotheses. Regarding the inclusion of additional levels of PB, we would like to point out that our simulations are already enormously time consuming in their present form. 

Minor issues:

Page 5: Not all QRPs are related to running multiple analyses and reporting only the smallest p-value. For example HARKing, rounding off p-values, or fraud. Furthermore, why did you choose these four? And do you expect they have exactly the same effect on the meta-analytical results, or not? What do we already know from previous studies on QRPs on for example effect size estimates?

Our response: We replaced the expression QRPs with the more apt “p-hacking” throughout. We selected our four particular types of p-hacking based on their inferred high prevalence (John, Loewenstein, & Prelec, 2012), see Line 112. To differentiate the impact of different forms of p-hacking is beyond the scope of our paper, and we are not aware of previous investigations of this question.

Page 6: Do we know how often 2-tailed publication bias is plausible, compared to one tailed bias? Is there some data from empirical studies?

Our response: We are not aware of data that would shed light on this question.

Page 9, line 189: please provide a reference for the claim that meta-analyses on continues outcomes are frequent.

Our response: We now provide a reference, van Erp, Verhagen, Grasman, and Wagenmakers (2017), see Line 198. The paper, which surveyed 705 MAs in Psychological Bulletin, does not directly mention types of outcome variables, but their open data show that >95% of MAs used Pearson’s r or a standardised mean difference as an effect size.

Page 9, line 201: Is this median N of 100 the total N or per group?

Our response: This is total N, which we now make clearer (see Line 201).

Page 12, line 271: An I2 value of 6.6% when true tau=0, in conditions without qrps or bias. This deviates much more from 0 than I would expect?

Our response: We now report the median (0.0) in addition to the mean (Line 285). Note that in the absence of true heterogeneity, tau can only be overestimated but not underestimated, which biases the mean.

Page 13, start of results: Please provide the reader with a sense of what the meta-analytical datasets looked like in the end: what was the effect of all qrps (splitting datasets, adding participants), on the actual sample sizes of primary studies? Is this still close to Ni=100?

Our response: This information is now added in L307. P-hacking increased mean sample size in primary studies only moderately.

Typos:

Page 5, line 111: QRPs instead of QRPS

Page 10, line 220: sections are not labeled as 2.2.

Page 13, line 288: URL to osf page no longer works. Please update the URL.

Page 15, line 347: ‘low k low’.

Page 24, line 500: ‘were’ instead of ‘where’.

Our response: Thanks, corrected.

Reviewer #2: “Heterogeneity estimates in a biased world” is a Monte Carlo study of the effects of publication bias (PB) and QRP (questionable research practices). It seems to be rigorously conducted, and its simulations are based on realistic research conditions as seen in psychology. Its major findings is: “Our results showed that biases in primary studies caused much greater problems for the estimation of effect size than for the estimation of heterogeneity.” This is an important lesson that meta-analysis community needs to hear. I suspect that this was already widely known, but I believe that this is the first paper that demonstrates this is a clear, rigorous and replicable way.

I wish to congratulate the authors for the way they conduct their Monte Carlo simulations. The design of the simulations can make an enormous difference to their results. Unless, the important research parameters (sample sizes, the amount of heterogeneity, the degree of publication selection, etc.) accurately reflects what is seen in the actual relevant research literature, the findings will be largely irrelevant. However, the authors based their simulations on what they found in what seems to be a fairly representative sample of 150 meta-analyses in psychology. I recommend that PLOS published this paper with a few revisions.

Our response: Thanks for your comments, from which we have learned a lot. Sorry that we failed to notice your review when we prepared our first revision. Please note that all line numbers in our reply refer to the clean copy, i.e., the copy without tracked changes.

Suggestions for revisions.

1. Emphasize main findings: Please emphasize and expand the main finding that it is PB and QRP that causes random effect (RE) to be so very bias, and that this bias is very large under the typical conditions that the authors simulate. This substantial bias has also been confirmed in a systematic review of large pre-registered multi-lab replications (Kvarven et al., 2020), and it is so large that RE is entirely unreliable if applied to psychology naively without many qualifications and auxiliary statistical checks. These biases are also of a notable scientific size. The authors need to state a bit more strongly how these different methods of estimating tau (the heterogeneity SD) are largely irrelevant, especially relative to the size and consequences of RE’s bias. These consequences need to be explicitly stated and emphasized.

Our response: In our discussion, we now place more emphasis on this point and reference some of the papers you indicated under (4): “In our simulation, biased primary studies caused much more severe problems for estimates of effect size than for estimates of heterogeneity. Therefore, future investigations into meta-analytic parameter estimations should prioritise how to deal with biases in effect size estimates (e.g., Duval & Tweedie, 2000; Egger, Smith, Schneider, & Minder, 1997; Henmi, Hattori, & Friede, 2021; Simonsohn, Nelson, & Simmons, 2014; Stanley, Doucouliagos, & Ioannidis, 2017; Stanley, Doucouliagos, Ioannidis, & Carter, 2021) over the relative merits of different heterogeneity estimators.” Lines 564-568

2. Biased studies: The way the authors characterize PB and QRP is rather misleading and may give the broader audience the wrong impression about the nature and extent of the problems involved. Classical PB is itself often interpreted as merely omitting some studies that are not statistically significant (SS). While this is indeed one avenue for the bias that we often find in published research results, there are many others. Reporting bias is recognized as a different avenue by medical researchers, as QRP is recognized by psychologists. But all of these vectors of the biases are the result of some process of selection of the results to be SS. This selection can be undertaken by the researchers on their own for their own reasons, or in anticipation of what reviewers and editors might demand. Or, this selection can be forced by the reviewers and editors. These details of selection process are largely irrelevant because they have the same outcome and can be simulated in the same way. Thus, a little more discussion of what this bias is and more references to the classical and better regraded methods to correct for these biases (collectively called PB here, for short) is needed. The authors repeatedly characterize this problem as “biases in primary studies.” It is not, or at least, this is not necessarily a bias in the primary studies. PB can be very serious, just as we see it in practice, if the individual primary results are not biased, but merely were selectively reported to be SS from entirely randomly produced distribution of estimates (with random QRPs, random outcome measures, random samples, etc). You might say that PB is an emergent property (selection for SS) of the entire research literature in a given area but is not associated with individually biased studies. Studies and researchers may also be biased, which will only amplify PB, but focusing on the unnecessary bias of individual studies can cause many to dismiss this severe problem. Many researchers do not believe that a notable portion of their colleagues are dishonest or deliberately distorting science. This is why, PB is so pernicious and easily dismissed. It can emerge from the system, as a whole, without individually researchers knowingly distorting science. Please do not characterize PB as ‘biased studies’ but rather as studies selected to be SS.

Our response: Wherever appropriate, we changed“biased primary studies” or similar to “biased sets of primary studies” or similar.

3. Type I errors: Please report the type I errors of all of these methods using the current simulation design. I suspect that the authors will find that RE has very high rates of type I errors for all of these methods, at least as long as there are more than a few estimates. If so, this will confirm the systematic review of large pre-registered multi-lab replications (Kvarven et al., 2020). Rates of false positives are very important as an indicator of scientific credibility. I suspect that RE (regardless of the method use to estimate tau) has such high rates of false positives, using the authors’ current simulation design, to disqualify RE from any serious scientific use. In any case, type I errors are important to show and to discuss. Not reporting type I errors could be considered to be a type of a selection bias in the way these simulation results are displayed and published. Methods PB, if you will.

Our response: We now address type-1 errors for mean effect size estimates in the new Fig9. We write “For overall effect size estimates (d), meta-analyses typically report a p-value, which is tacitly assumed to provide an appropriate safeguard against type-1 errors. Fig9 shows type-1 error rates for d under 1-tailed publication bias in our simulation. (Under 2-tailed publication bias, type-1 error rates proved very close to the nominal 5%.) As can be seen, type-1 error rates might reach catastrophic levels. Random effects p-values for d will therefore fail to offer protection against type-1 errors unless publication bias and p-hacking can be ruled out.” LL512-517.

4. Alternatives to random effects: This entire study assumes that RE is the only adequate method to conduct basic meta-analysis in psychology and that this issue then comes down to the best way to calculate RE. This is not the case, and worse, the authors show that all the ways to calculate RE produce notably large bias (greatly exaggerating the size of the effect under examination). I suspect that this simulation design will show that RE has high rates of false positives. It has long been known that RE has unacceptable biases and that these biases are easily reduced (Henmi and Copas, 2010; Stanley and Doucouliagos, 2015). Henmi and Copas (2010) showed that FE (fixed effect) will notably reduces PB and that the RE’s estimate of tau can accommodate the heterogeneity that FE ignores. However, Henmi and Copas (2010) uses the DL estimate of tau in their calculation of the CI. So, the estimate of tau might still be important in their approach. Henmi and others (2021) has recently generalized this method and show how it can work for very small meat-analyses. Alternatively, an entirely different approach, the unrestricted weighted least squares (UWLS), uses the bias reduction of FE but automatically accommodates heterogeneity using the mathematical invariance of WLS’s variance-covariance matrix to any multiplicative constant. UWLS accommodates heterogeneity without referring to or using RE or any of its estimates of tau (Stanley and Doucouliagos, 2015; 2017). That is, the central issue of this study of the effect of PB on estimates of tau could be entirely avoided and, at the same time, reduce the large biases reported in this paper. Simulations, like these, have shown that UWLS notably reduces RE’s bias with little if any compensating statistical loss (Stanley and Doucouliagos, 2015; 2017). These alternative methods to RE have been widely applied across the disciplines and used as a basis for a new statistical method to detect PB (Stanley et al., 2021). It would be nice if these other methods were simulated and reported using this same design. At a minimum discussed, they need to be discussed as viable alternative to this concern about how tau is calculated and as an alternative to RE’s large biases and high rates of false positives. The central scientific question, is how to reduce or eliminate bias and false positive meta-analyses because they are often the best scientific evidence we have.

Our response: We now reference some of these papers. (See our reply to your first comment for details.) In the context of UWSL, you write “the central issue of this study of the effect of PB on estimates of tau could be entirely avoided”. This is true if the aim of the meta-analysis is restricted to estimating the average effect size with an appropriate CI. However, as we point out in the section Why heterogeneity matters, the extent of effect size heterogeneity can be of substantial interest in itself. For example, it might be applied to understand which psychological effects change most and least across cultures. As heterogeneity is at the core of our paper, we certainly cannot avoid it. We agree that extensions of our simulations to models beyond RE are of interest. However, any research project must be limited, and we decided to focus on the (still very popular) RE model here.

References:

Henmi M, Copas JB. Confidence intervals for random effects meta-analysis and robustness to publication bias. Statistics in Medicine, 2010; 29:2969–2983.

Henmi M, Hattori S, Friede T. A confidence interval robust to publication bias for random-effects meta-analysis of few studies. Res Syn Meth. 2021;12:674–679. https://doi.org/10.1002/jrsm.1482

Stanley, T.D. and Doucouliagos, C. Neither fixed nor random: Weighted least squares meta-analysis,” Statistics in Medicine, 2015: 342116-27.

Stanley, T.D. and Doucouliagos, C. Neither fixed nor random: Weighted least squares meta-regression analysis. Res Synth Methods. 2017;8:19-42.

Stanley TD, Doucouliagos H, Ioannidis JPA, Carter EC. Detecting publication selection bias through excess statistical significance. Research Synthesis Methods. 2021; 1-20. https://doi.org/10.1002/jrsm.1512

Reviewer #3: Review comments to the author can be found in the attached .docx document. They are organised in the sequence of the paper and include some general points and more specific questions to be responded to.

6. PLOS authors have the option to publish the peer review history of their article (what does this mean?). If published, this will include your full peer review and any attached files.

Do you want your identity to be public for this peer review? For information about this choice, including consent withdrawal, please see our Privacy Policy.

Reviewer #1: Yes: Hilde Augusteijn

Reviewer #2: No

Reviewer #3: No

Our response:

Augusteijn, H. E., van Aert, R., & van Assen, M. A. (2019). The effect of publication bias on the Q test and assessment of heterogeneity. Psychological Methods, 24(1), 116-134. 

Duval, S., & Tweedie, R. (2000). Trim and fill: A simple funnel‐plot–based method of testing and adjusting for publication bias in meta‐analysis. Biometrics, 56(2), 455-463. 

Egger, M., Smith, G. D., Schneider, M., & Minder, C. (1997). Bias in meta-analysis detected by a simple, graphical test. BMJ, 315(7109), 629-634. 

Henmi, M., Hattori, S., & Friede, T. (2021). A confidence interval robust to publication bias for random‐effects meta‐analysis of few studies. Research synthesis methods. 

John, L. K., Loewenstein, G., & Prelec, D. (2012). Measuring the prevalence of questionable research practices with incentives for truth telling. Psychological science, 23(5), 524-532. 

Kühberger, A., Fritz, A., & Scherndl, T. (2014). Publication bias in psychology: a diagnosis based on the correlation between effect size and sample size. PloS one, 9(9), e105825. 

Levine, T. R., Asada, K. J., & Carpenter, C. (2009). Sample sizes and effect sizes are negatively correlated in meta-analyses: Evidence and implications of a publication bias against nonsignificant findings. Communication Monographs, 76(3), 286-302. 

Macnamara, B. N., Hambrick, D. Z., & Oswald, F. L. (2014). Deliberate practice and performance in music, games, sports, education, and professions: A meta-analysis. Psychological science, 25(8), 1608-1618. 

Simonsohn, U., Nelson, L. D., & Simmons, J. P. (2014). P-curve: a key to the file-drawer. Journal of experimental psychology: General, 143(2), 534-547. 

Sisk, V. F., Burgoyne, A. P., Sun, J., Butler, J. L., & Macnamara, B. N. (2018). To what extent and under which circumstances are growth mind-sets important to academic achievement? Two meta-analyses. Psychological science, 29(4), 549-571. 

Stanley, T., Doucouliagos, H., & Ioannidis, J. P. (2017). Finding the power to reduce publication bias. Statistics in Medicine, 36(10), 1580-1598. 

Stanley, T., Doucouliagos, H., Ioannidis, J. P., & Carter, E. C. (2021). Detecting publication selection bias through excess statistical significance. Research synthesis methods, 12, 776–795. 

van Erp, S., Verhagen, J., Grasman, R. P., & Wagenmakers, E.-J. (2017). Estimates of between-study heterogeneity for 705 meta-analyses reported in Psychological Bulletin from 1990–2013. Journal of Open Psychology Data, 5(1).

---

## [Decision Letter · Decision Letter 2]

6 Jan 2022

Heterogeneity estimates in a biased world

PONE-D-21-24961R2

Dear Dr. Hönekopp,

We’re pleased to inform you that your manuscript has been judged scientifically suitable for publication and will be formally accepted for publication once it meets all outstanding technical requirements.

Kind regards,

Tim Mathes

Academic Editor

PLOS ONE

Additional Editor Comments (optional):

Reviewers' comments:

Reviewer's Responses to Questions

**Comments to the Author**

1. If the authors have adequately addressed your comments raised in a previous round of review and you feel that this manuscript is now acceptable for publication, you may indicate that here to bypass the “Comments to the Author” section, enter your conflict of interest statement in the “Confidential to Editor” section, and submit your "Accept" recommendation.

Reviewer #2: All comments have been addressed

2. Is the manuscript technically sound, and do the data support the conclusions?

Reviewer #2: Yes

3. Has the statistical analysis been performed appropriately and rigorously? 

Reviewer #2: Yes

4. Have the authors made all data underlying the findings in their manuscript fully available?

Reviewer #2: (No Response)

5. Is the manuscript presented in an intelligible fashion and written in standard English?

Reviewer #2: Yes

6. Review Comments to the Author

Reviewer #2: (No Response)

7. PLOS authors have the option to publish the peer review history of their article (what does this mean?). If published, this will include your full peer review and any attached files.

Reviewer #2: No

---

## [Editor Report · Acceptance letter]

18 Jan 2022

PONE-D-21-24961R2 

Heterogeneity estimates in a biased world 

Dear Dr. Hönekopp:

I'm pleased to inform you that your manuscript has been deemed suitable for publication in PLOS ONE. Congratulations! Your manuscript is now with our production department. 

Kind regards, 

on behalf of

Dr. Tim Mathes 

Academic Editor

PLOS ONE